# Plug-and-play metabolic transducers expand the chemical detection space of cell-free biosensors

Peter L. Voyvodic[1], Amir Pandi [2], Mathilde Koch[2], Ismael Conejero[1,3,4], Emmanuel Valjent [5], Philippe Courtet [3,6], Eric Renard[5,7], Jean-Loup Faulon[2,8,9] & Jerome Bonnet [1,9]

Cell-free transcription–translation systems have great potential for biosensing, yet the range of detectable chemicals is limited. Here we provide a workflow to expand the range of molecules detectable by cell-free biosensors through combining synthetic metabolic cascades with transcription factor-based networks. These hybrid cell-free biosensors have a fast response time, strong signal response, and a high dynamic range. In addition, they are capable of functioning in a variety of complex media, including commercial beverages and human urine, in which they can be used to detect clinically relevant concentrations of small molecules. This work provides a foundation to engineer modular cell-free biosensors tailored for many applications.

[1] Centre de Biochimie Structurale, INSERM U1054, CNRS UMR5048, University of Montpellier, Montpellier 34090, France. [2] Micalis Institute, INRA, AgroParisTech, Université Paris-Saclay, Jouy-en-Josas 78352, France. [3] INSERM U1061, Neuropsychiatry, Epidemiological and Clinical Research, Montpellier 34093, France. [4] Department of Psychiatry, Caremeau Hospital, University Hospital of Nîmes, Nîmes 30900, France. [5] UMR CNRS 5203/INSERM U1191, Institute of Functional Genomics, University of Montpellier, Montpellier 34090, France. [6] CHU Montpellier, Department of Emergency Psychiatry and Post Acute Care, Lapeyronie Hospital, CHRU Montpellier, Montpellier 34090, France. [7] Department of Endocrinology, Diabetes, Nutrition and CIC INSERM 1411, Montpellier University Hospital, Montpellier 34295, France. [8] SYNBIOCHEM Center, School of Chemistry, University of Manchester, Manchester M1 7DN, UK. [9] These authors contributed equally: Jean-Loup Faulon, Jerome Bonnet. Correspondence and requests for materials should be addressed to J.-L.F. (email: jean-loup.faulon@inra.fr) or to J.B. (email: jerome.bonnet@inserm.fr)

There is currently an urgent need for low-cost biosensors in a variety of fields from environmental remediation to clinical diagnostics[1–3]. The ability of living organisms to detect signals in their environment and transduce them into a response can be used to create cheap, novel sensors with high sensitivity and specificity. By leveraging the ability of transcription factors (TFs) to control gene expression, synthetic biologists have genetically engineered microbes to detect a wide range of compounds, from clinical biomarkers to environmental pollutants[4–7].

Cell-free transcription/translation (TXTL) systems have great promise as the next generation of synthetic biology-derived biosensors. They are cheap to produce[8], abiotic, and can be lyophilized such that they are stable at room temperature for up to one year: a vital necessity for point-of-care applications such as low-resource nation and home diagnostic use[9]. Cell-free TXTL toolboxes have been designed that support the operation of many of the circuits previously engineered in vivo[10,11]. Encapsulated cell extracts can also be used in combination with living cells to produce new sensing modalities[12]. Cell-free biosensors were engineered to successfully detect Zika virus in rhesus macaques and an acyl homoserine lactone, 3OC12-HSL, from *Pseudomonas aeruginosa* in human clinical samples[13,14]. However, current cell-free biosensors have been limited to detection of nucleic acid sequences, via toehold displacement, or well-characterized TF ligands.

Here we put forward a generalized, modular workflow using metabolic transducers to rapidly expand the chemical space detectable by cell-free biosensors in a plug-and-play manner. We then illustrate our workflow with a proof-of-concept example: the transcription factor BenR, which is activated by benzoic acid, and two metabolic modules HipO and CocE, which convert hippuric acid and cocaine, respectively, into benzoic acid. Each component is individually cloned into a cell-free vector, such that the DNA concentrations can be titrated over three orders of magnitude to optimize sensor performance. Finally, we demonstrate that these sensors can function in complex solutions, detecting benzoic acid in commercial beverages and hippuric acid and cocaine in human urine.

## Results

**Design workflow for cell-free biosensors.** Synthetic metabolic cascades have been used by the synthetic biology community for a wide range of applications, including production of biofuels, pharmaceuticals, and biomaterials[15–17]. As such, there is a wide variety of well-characterized enzymes catalyzing various reactions transforming one molecule into another. Our framework harnesses this power by using metabolic enzymes as transducers to allow us to "plug in" a given enzyme into our characterized biosensor modules to detect a ligand with no known corresponding TF (Fig. 1a). Specifically, the metabolic enzyme converts the undetectable molecule into one for which we have an existing TF-based genetic circuit (Fig. 1b). We used the SensiPath webserver that we previously designed and validated in vivo to determine the required metabolic cascade[18,19] (Supplementary Note 1).

The workflow to engineer a cell-free biosensor detecting a novel molecule is straightforward (Fig. 1c). First, possible metabolic pathways to convert the molecule of interest into a detectable ligand are identified using SensiPath. Second, the genes coding for the metabolic transducer enzyme, the TF sensor, and the reporter module are synthesized and cloned into cell-free expression vectors. Finally, the DNA concentration of each plasmid is titrated in cell-free reactions to optimize signal strength and dynamic range in response to the molecule of interest (Fig. 1c).

**Optimization of cell-free benzoic acid sensor.** BenR is a member of the AraC/XylS TF family, originally isolated from *Pseudomonas putida*. In the presence of benzoate, BenR binds to the $P_{Ben}$ promoter and activates transcription (Fig. 2a). To engineer a cell-free biosensor detecting benzoate, we cloned BenR under the control of the OR2-OR1-Pr promoter, a modified version of the lambda phage repressor promoter Cro, known to express strongly in cell-free systems[20]. The $P_{Ben}$ promoter driving superfolder green fluorescent protein (sfGFP) was cloned in a separate plasmid. After initial pilot tests demonstrated that BenR was functional in a cell-free environment, we optimized the BenR biosensor by titrating the DNA concentration of the TF and reporter plasmids.

One advantage of working in a cell-free framework is that the DNA concentration is directly controlled by pipetting. As such, the process of finding an optimal DNA concentration is relatively straightforward: we created a matrix of DNA concentrations for TF and reporter plasmids between 0 and 100 nM, and induced these different cell-free reactions using four different concentrations of benzoic acid: 0, 10, 100, and 1000 μM (Fig. 2b and Supplementary Table 1).

Encouragingly, the system had extremely low background signal in the absence of benzoic acid, indicating that the $P_{Ben}$ promoter has very little "leakiness" in a cell-free environment. When benzoic acid was added to the reaction, the sfGFP output signal was clearly detectable and fluorescence intensity was correlated with increasing reporter plasmid concentration. However, the signal reached a plateau for increasing concentrations of TF plasmid at 30 nM. We hypothesize that this plateau is due to competition for transcriptional and translational resources between TF and reporter plasmids. This plateau is also observed in a mathematical model of cell-free biosensors (Supplementary Notes 2-3 and Supplementary Fig. 1). Based on these data, we set the optimal plasmids concentrations to 30 nM for the TF plasmid and 100 nM for the reporter plasmid.

Compared with its in vivo counterpart[18], the cell-free benzoic acid biosensor is faster (maximum signal reached in 4 h, Supplementary Fig. 2), has a much higher sensitivity and dynamic range, and has a maximum fold change of over 200 (vs. ~10-fold in vivo) (Fig. 2c). These results exemplify the advantages of cell-free systems for rapidly engineering biosensors with optimal properties.

**Expanding the detection space of the benzoic acid sensor with metabolic transducers.** With the sensor and output modules optimized, we demonstrated the ability of our system to expand its chemical detection space using different metabolic transducer modules. HipO is an enzyme from *Campylobacter jejuni* and CocE is an esterase from *Rhodococcus sp.* that convert hippuric acid and cocaine into benzoic acid, respectively. We cloned each enzyme into the cell-free expression vector and, using the optimized DNA concentrations of TF and reporter plasmids, titrated different concentrations of metabolic transducer DNA for a range of inducer inputs (Fig. 3a and Supplementary Table 2). Interestingly, we observed a clear peak in sfGFP signal corresponding to a particular concentration effectiveness: 3 nM for HipO and 10 nM for CocE. We built several mathematical models based on different assumptions that could reproduce the observed bell-shaped response to enzyme DNA concentration as well as its shift between the two enzymes (Supplementary Fig. 3). Based on these models, we hypothesized that the observed bell-shaped response is likely due to competition between the different modules, leading to an important and unnecessary enzyme production at high DNA concentrations that diverts resources such as RNA polymerase, ribosomes, and energy from sfGFP transcription and

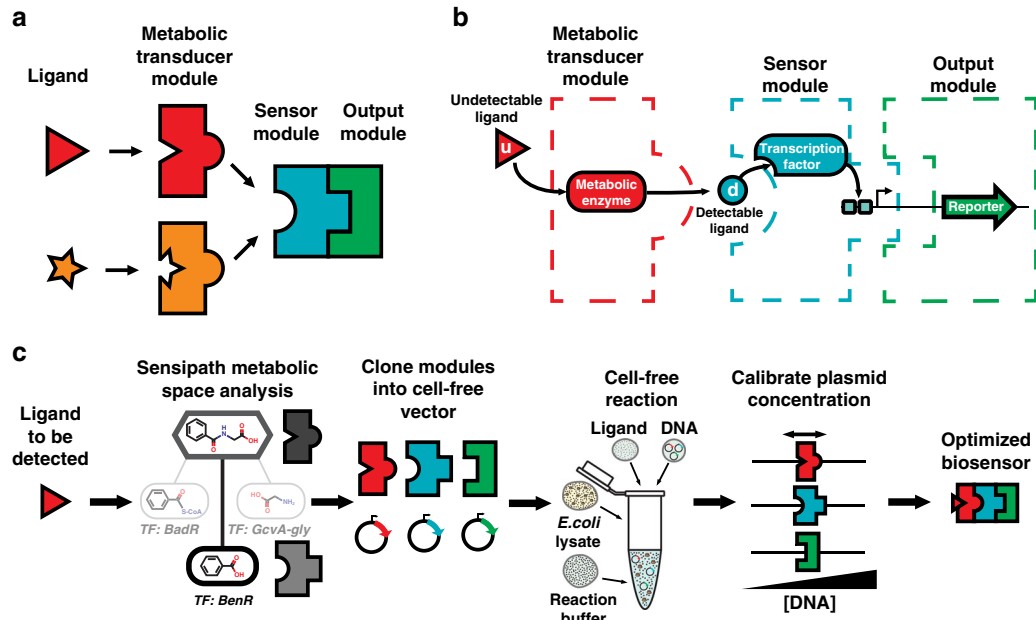

**Fig. 1** A modular design workflow for engineering scalable cell-free biosensors. **a** Cell-free biosensors are composed of three modules: a generic sensor module linked to an output module and a metabolic transducer module transforming different molecules into ligands detectable by the sensor module. **b** An undetectable ligand is converted into a detectable ligand by the enzyme from the transducer module. Binding to the transcription factor controls the sensor module and downstream gene expression. **c** The biosensor design workflow starts with retrosynthetic pathway design using the SensiPath server[19]. Once the transducer and sensor modules are determined, the genes encoding enzymes, transcription factors, and target promoters driving a reporter are cloned into cell-free expression vectors. The sensor is calibrated by titrating the concentrations of each plasmid to maximize signal output and dynamic range

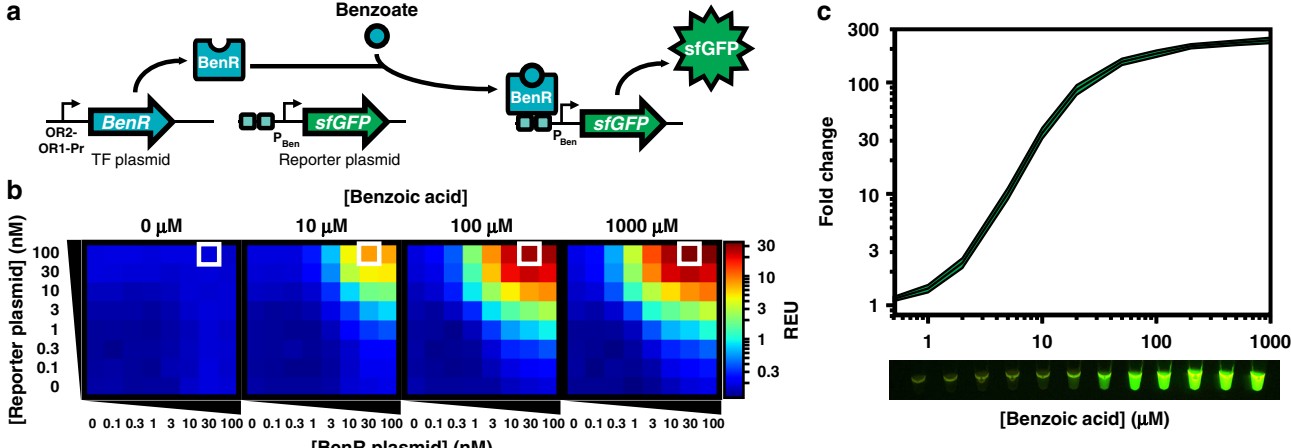

**Fig. 2** Calibration of sensor and output modules for benzoate detection. **a** BenR binds to the $P_{Ben}$ promoter in the presence of benzoate and activates gene expression. Here, BenR is cloned in the pBEAST plasmid (a derivative of pBEST[20]) and driven by a strong constitutive promoter, OR2-OR1-Pr. The $P_{Ben}$ promoter is cloned into another pBEAST backbone and drives expression of superfolder green fluorescent protein (sfGFP). As the system operates without a cellular boundary, multiple plasmids encoding different components of the network can easily be used simultaneously. Plasmid concentrations can then be fine tuned to identify optimal operating conditions. **b** Optimization of the BenR sensor and reporter modules. Cell-free reactions of 20 μl containing different concentrations of the BenR and reporter plasmids were prepared and their response to different concentrations of benzoic acid were monitored. The white square represents the optimal condition (100 nM reporter and 30 nM BenR plasmid) with the highest relative fluorescence (see Supplementary Fig. 2 and Supplementary Table 1). Reactions were run in sealed 384-well plates in a plate reader at 37 °C for at least 8 h. The heat maps represent the signal intensity after 4 h. Data are the mean of three experiments performed on three different days and all fluorescence values are expressed in Relative Expression Units (REU) compared with 100 pM of a strong, constitutive sfGFP-producing plasmid. See Methods for more details. **c** Upper panel: The BenR sensor can detect benzoic acid over three orders of magnitude and at concentrations as low as 1 μM. Shaded area around curves corresponds to ±SEM of the three experiments. Lower panel: GFP expression in response to the same range of concentrations of benzoic acid as in the upper panel is easily detectable by eye on a UV table. Source Data are available in the Source Data File

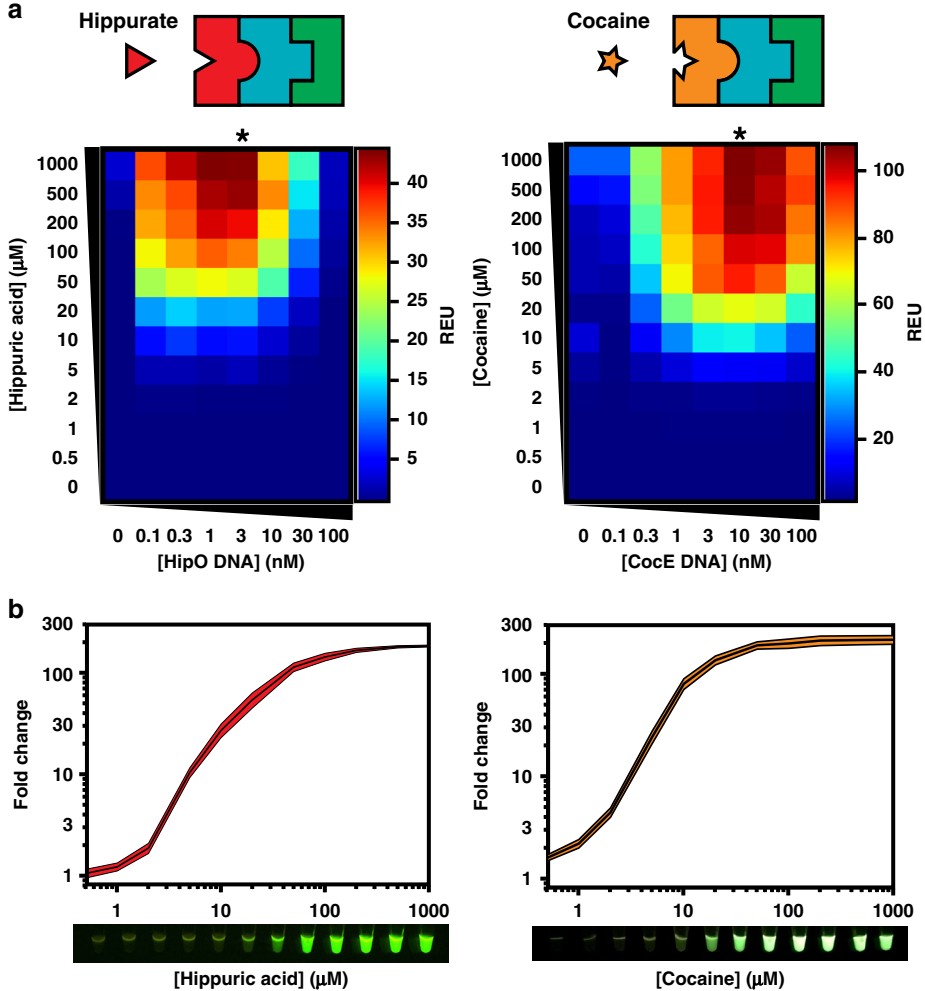

**Fig. 3** Metabolic transducers expand the chemical detection space of cell-free biosensors. **a** Hippurate or cocaine can be detected using different metabolic transducers. Plasmids encoding the HipO or CocE enzymes, which convert hippuric acid or cocaine into benzoic acid, were mixed at different concentrations with optimal BenR and reporter plasmids concentrations as determined in Fig. 2 (30 nM and 100 nM, respectively). These reactions were then incubated with increasing concentrations of inducer for at least 8 h. The heat maps represent the signal intensity after four hours (Supplementary Figs. 6-7 and Supplementary Table 2). Asterisks denote the optimal DNA concentration for the metabolic module. Data are the average of three experiments performed on three different days and all fluorescence values are expressed in Relative Expression Units (REU) compared with 100 pM of a strong, constitutive sfGFP-producing plasmid. **b** Optimized cell-free biosensors incorporating a metabolic transducer module exhibit comparable performance to the BenR sensor module (from Fig. 2c). All data are the mean of three experiments performed on three different days. Shaded area around curves corresponds to ±SEM of the three experiments. See Methods for more details. Lower panel: GFP expression in cell-free reactions in response to various concentrations of inducer visualized on a UV table. Source Data are available in the Source Data File

translation, as well as generates toxic byproducts. Moreover, we provide evidence that the shifting peak between the two setups is most likely due to lower expression of CocE (Supplementary Notes 2-3 and Supplementary Fig. 4). In addition, the model hypothesized that using a higher TF concentration would necessitate a higher level of metabolic enzyme without an increase in overall signal, a shift that we subsequently observed experimentally (Supplementary Notes 2-3 and Supplementary Fig. 5).

A key observation is that even at high levels of inducer, there is very little signal in the absence of DNA encoding the metabolic transducer. These data indicate that the metabolic enzyme is essential for sensor selectivity and differentiation between hippuric acid and cocaine from benzoic acid, and that both molecules have minimal off-target binding to BenR. Strikingly, both the hippuric acid and cocaine biosensors exhibit fold change and detection range highly similar to that of the benzoic acid sensor, demonstrating the high conversion rate of the metabolic transducer (Fig. 3b). The conversion also appears to be extremely

fast, as no significant difference was observed in response kinetics with or without the metabolic transducer, although the lower incubation temperature of the cocaine biosensor showed slightly slower kinetics (Supplementary Figs. 2, 6, and 7).

**Detection of benzoic acid, hippuric acid, and cocaine in complex samples.** Although the results of our optimized biosensors were promising, the intended final environment in which they should operate is far more complex. We thus sought to test their capabilities for real-world applications. Benzoic acid and sodium benzoate are widely used food additives for preservation. Although classified as Generalized Recognized As Safe by the United States Food and Drug Administration, their maximal levels in foodstuffs are limited to 0.1%. In addition, some people respond poorly to their consumption, particularly patients suffering from chronic inflammation or orofacial granulomatosis, who are frequently placed on benzoate-free diets by their physicians[21,22]. Lastly, there is evidence that when benzoates are

added to beverages in the presence of ascorbic acid, they can be converted into low levels of benzene, a strong carcinogen[23,24]; this reaction is enhanced by increased temperatures, which frequently occur during transportation. In this context, a simple assay for detecting benzoic acid could be useful.

To test whether our benzoic acid sensor could function as a monitor in the food industry, we procured several different drinks from a local supermarket. The nutritional information of each beverage included benzoic acid, sodium benzoate, or no benzoates. Strikingly, after adding 2 μL of the beverages directly to 20 μL reactions of our optimized benzoic acid sensor, we were able to distinguish which beverages contained benzoates with 100% accuracy after only 1 h of incubation (Fig. 4a and Supplementary Fig. 8). The beverages were composed of two categories: carbonated orange drinks and Monster® energy drinks. Despite similarities between the non-benzoate ingredients in each class, our cell-free benzoic acid biosensor rapidly produced sfGFP in beverages that listed benzoate ingredients with fold changes up to ~180.

Although our system has the ability to quickly detect benzoates by directly adding the beverages to the reaction, we noticed that there was up to 75% inhibition to some of the cell-free reactions when comparing expression of a constitutive promoter to a control (Supplementary Fig. 9). Therefore, to test our sensor's ability to quantify benzoates, we performed an experiment with a 1:10 dilution, which showed minimal reaction interference (Supplementary Fig. 9), and converted the resulting fluorescence intensities to concentrations using a calibration curve from a benzoic acid standard (Supplementary Fig. 10). These results were compared against measurements from liquid chromatography-mass spectrometry (LC-MS) (Fig. 4b and Supplementary Table 3). Seven of the ten drinks showed very strong agreement between the quantitative results from our sensor and the LC-MS results. Three of the beverages (Monster® Absolutely Zero, Monster® Ultra, and Monster® Ultra Red) had diminished cell-free values relative to those from LC-MS. Taken together, these results demonstrate that our sensors can remain functional in commercial products and rapidly detect and quantify benzoates.

We then wanted to test whether our hippuric acid sensor could detect endogenous levels of a molecule of interest in a clinical context. Hippuric acid has long been known to be regularly excreted by humans in urine as the end product of several different aromatic compounds, including benzoates, which are converted in the liver[25]. Hippuric acid has been correlated with high levels of toluene exposure in some operational conditions[26]. In addition, recent research by Isabella et al.[27] used hippuric acid as a biomarker for monitoring the efficacy of treatment against phenylketonuria, a neurotoxic disease characterized by the inability to process the amino acid phenylalanine. We thus wanted to test whether our sensor could detect clinical levels of endogenous hippuric acid in human urine. When adding 2 μL of a 1:10 dilution to a 20 μL reaction (1% cell-free reaction concentration), we found little interference from urine to expression of a constitutive GFP plasmid relative to the positive control (Supplementary Fig. 11). When testing the urine for hippuric acid, we observed little to no response from our benzoic acid sensor (without the HipO-expressing plasmid) (Supplementary Table 4), but the complete hippuric acid sensor gave levels that fell within our calibration curve (Supplementary Fig. 12). Urinary hippuric acid concentrations estimated using our cell-free biosensor closely matched the values determined by LC-MS ($R^2 = 0.98$; Supplementary Fig. 13, Fig. 4c, and Supplementary Table 5). These data are a promising step toward developing cell-free biosensors for biomarker detection in clinical samples.

Finally, we aimed to detect cocaine in clinically relevant conditions. Cocaine rapidly enters the bloodstream after ingestion

and is subsequently detectable in the urine for up to 10 h[28]. To determine whether our system could detect clinically relevant cocaine levels, we spiked urine samples with increasing concentrations of cocaine and added 2 μL to 20 μL cell-free reactions with our optimized cocaine biosensor. Our initial experiment showed small but detectable sfGFP signal at urinary concentration of 1000 μM, but our system was unable to show adequate fold change at lower, clinically relevant concentrations (Supplementary Fig. 14). We found that cell-free reactions produce increasing low levels of noise over time in the GFP fluorescence channel (Supplementary Fig. 15) and hypothesized that we could increase our signal-to-noise ratio by changing our reporter to luciferase. We cloned the firefly luciferase gene under control of the $P_{Ben}$ promoter and in an initial test indeed observed an increase in signal-to-noise ratio (Supplementary Fig. 16). We then added increasing cocaine concentrations into six different samples containing our cell-free cocaine sensor with the luciferase reporter (Fig. 4d). Five of the six sample showed strong fold change, with detectable fold changes of 4.3–8.8 at previous clinically detected cocaine concentrations in urine[29]. One sample (U3) showed minimal fold change due to high background signal that was also observed using the benzoic acid sensor (Supplementary Fig. 17). As the urine samples were supplied by subjects from the endocrinology department, it is possible that the medical condition of this patient resulted in interfering metabolites in the urine that can activate the BenR system. This background signal was minimal when we detected hippuric acid in urine, likely because of the urine samples dilution step (Supplementary Table 4). In conclusion, these data demonstrate that our cell-free biosensors can be used to detect clinically relevant levels of drugs and endogenous metabolites in pure, unprocessed clinical samples.

## Discussion

This work demonstrates that we can engineer modular, cell-free biosensors that can be easily calibrated to have high signal strength and dynamic range and can function in complex detection environments. Upon engineering a cell-free biosensor for benzoic acid, we show that the system can be scaled by using different metabolic transducer modules to expand the chemical space that each sensor/reporter pair can detect. In addition, we provide a three order-of-magnitude titration for each DNA component to optimize cell-free biosensor performance along with a mathematical model enabling a better understanding of the parameters governing cell-free biosensor response, which will help future optimization of such devices. By demonstrating that these sensors can function in samples from the food and beverage industry, as well as complex clinical samples such as human urine, we provide an example for their potential outside the lab in real-world applications. This is the first time, to our knowledge, that cell-free biosensors have been used to detect endogenous molecules in unprocessed samples.

Using our workflow, this process should be applicable to a wide range of other sensor/reporter pairs. One constraint of our system is that the TF must respond only to the product of the enzymatic reaction and not the substrate. Such potential crosstalk can easily be checked by running a control reaction without the metabolic transducer module. We computed that 1205 disease-associated biomarkers from the Human Metabolome Database (HMDB) could be converted into detectable molecules by one enzymatic reaction (Supplementary Note 1 and Supplementary Data 1). In addition, 64 HMDB metabolites could be transformed into benzoate and thus theoretically connected via a metabolic transducer to our optimized sensor (Supplementary Note 1 and Supplementary Data 2).

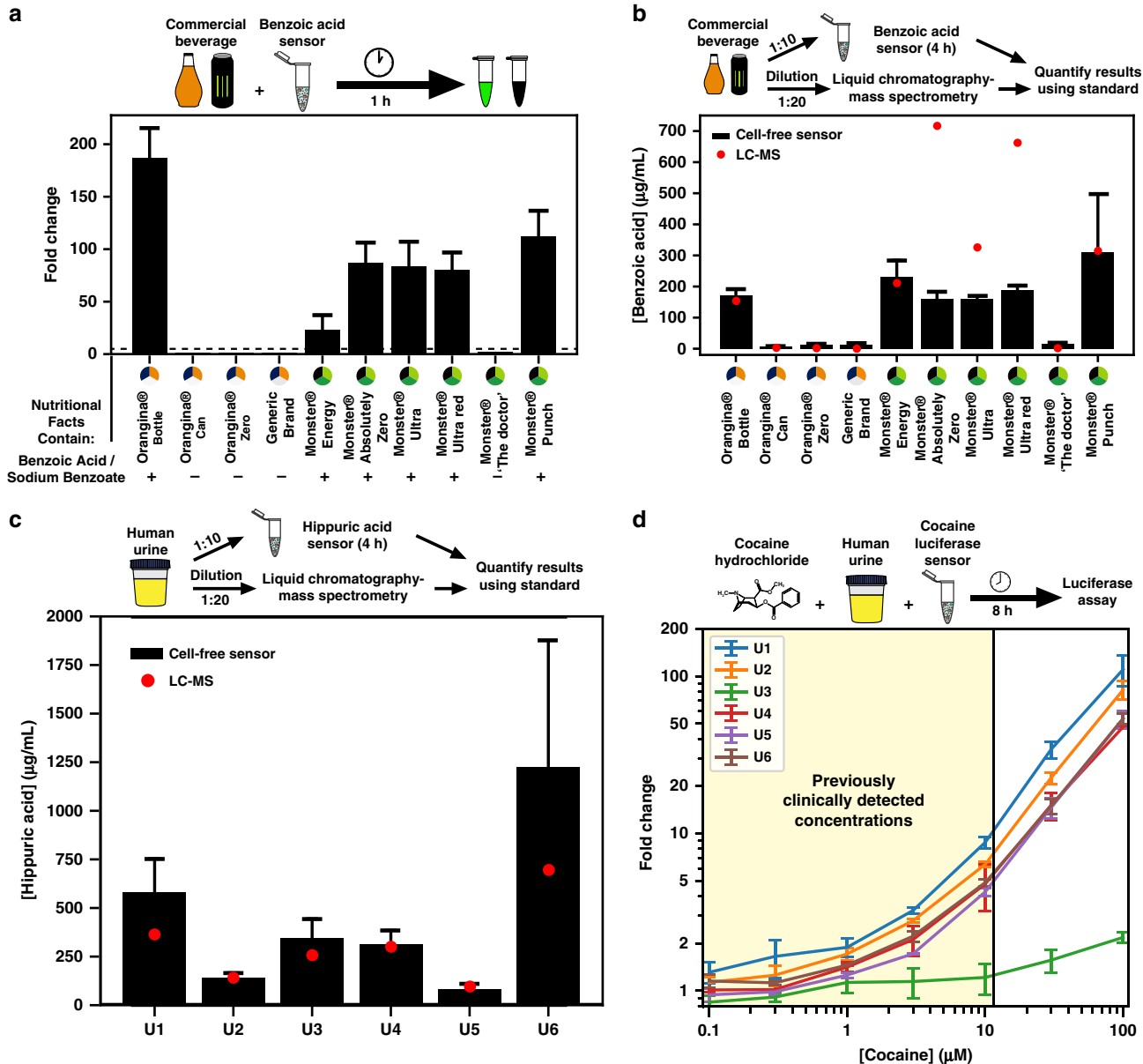

**Fig. 4** Detecting benzoic acid, hippuric acid, and cocaine in complex samples. **a** Cell-free benzoic acid sensor can detect benzoates in commercial beverages. Addition of an array of different orange and energy drinks to the optimized benzoic acid biosensor produces up to ~180 fold-change response relative to the negative control with water after 1 h incubation at 37 °C. The test showed 100% specificity and sensitivity to detection of benzoates based on their inclusion in the ingredient label using a fold change of 5 as the cut-off point (dashed line). **b** Benzoic acid sensor is capable of quantifying the concentration of benzoic acid in different beverages. Beverages were added at 1:10 dilution to cell-free reactions for 4 h and the benzoic acid concentration was determined using a calibration curve (Supplementary Fig. 10). Results were compared with those determined by LC-MS. **c** Endogenous hippuric acid in urine can be quantified with a cell-free biosensor. Clinical urine samples (U1–U6) were diluted 1:10 and added to the optimized hippuric acid sensor for four hours at 37 °C after which endogenous hippuric acid concentration was determined using a calibration curve (Supplementary Fig. 12). Results were compared with those determined by LC-MS. **d** Cocaine can be detected in clinical urine samples at previously clinically detected concentrations. Cocaine titrations were added to clinical human urine samples (U1–U6) and cell-free cocaine luciferase-output biosensors and incubated at 30 °C for 8 h. Subsequently, a luciferase assay was performed to determine the presence of cocaine. The colored region represents the concentration of cocaine previously measured in human clinical samples from hospitalized patients (40.13 μg/mL or 118 μM cocaine concentration in urine, corresponding to a 11.8 μM final concentration in the cell-free reaction—2 μL urine in a 20 μL reaction)[29]. All curves are plotted for the mean of three experiments performed on three different days. Error bars correspond to ±SD from the mean of the three experiments. See Methods for more details. Source Data are available in the Source Data File

Further improvements to our platform could include exploring sample pre-processing methods that could increase sensor robustness[30,31] together with adaptation into an off-the-shelf format more amenable to point-of-care applications[9,32]. Also, although we could detect clinically relevant concentrations of cocaine, this application would benefit from achieving a lower

detection limit, e.g., through the use of downstream genetic amplifiers[33].

In summary, by rapidly expanding the number of detectable compounds and remaining functional even in complex samples, cell-free biosensors using plug-and-play metabolic transducers could be used to address many challenges in environmental

detection, drug enforcement, and point-of-care medical diagnostics.

## Methods

**Molecular biology**. All clones were based on a previously characterized cell-free expression plasmid (pBEST-OR2-OR1-Pr-UTR1-deGFP-T500 was a gift from Vincent Noireaux [Addgene plasmid #40019][20]). To better facilitate cloning with a range of techniques and any future component insertion into larger gene circuits, the construct was modified by adding 40 base pair spacers and an upstream terminator, and renamed pBEAST. Clones were created via Gibson or Golden Gate assembly in DH5αZ1 chemically competent *Escherichia coli* where the deGFP was replaced by BenR or HipO. For CocE, the promoter was changed to another strong constitutive promoter, J23101, and the RBS was changed to B0032. The reporter plasmid for P_Ben used the native RBS from *P. putida* and sfGFP as the output, which was found to produce a strong signal with fast folding time in cell-free reactions at 37 °C. For experiments testing cocaine levels in urine, the sfGFP output was changed to firefly luciferase via Gibson assembly cloning. DNA for cell-free reactions was prepared from overnight bacterial cultures using Maxiprep kits (Macherey-Nagel).

**Extract preparation**. Cell-free *E. coli* extracts were produced using a modified version of existing protocols[8,34]. Briefly, an overnight culture of BL21 Star (DE3)::RF1-CBD₃*E. coli* was used to inoculate 660 mL of 2xYT-P media in each of six 2 L flasks at a dilution of 1:100. The cultures were grown at 37 °C with 220 r.p.m. shaking for ~3.5 h until the OD 600 = 2.0. Cultures were spun down at 5000 × g at 4 °C for 12 min. Cell pellets were washed twice with 200 mL S30A buffer (14 mM Mg-glutamate, 60 mM K-glutamate, 50 mM Tris pH 7.7), centrifuging afterwards at 5000 × g at 4 °C for 12 min. Cell pellets were then resuspended in 40 mL S30A buffer and transferred to pre-weighed 50 mL Falcon conical tubes where they were centrifuged twice at 2000 × g at 4 °C for 8 and 2 min, respectively, removing the supernatant after each. Finally, the tubes were reweighed and flash frozen in liquid nitrogen before storing at −80 °C.

Cell pellets were thawed on ice and resuspended in 1 mL S30A buffer per gram cell pellet. Cell suspensions were lysed via a single pass through a French press homogenizer (Avestin; Emulsiflex-C3) at 15,000–20,000 psi and then centrifuged at 12,000 × g at 4 °C for 30 min to separate out cellular cytoplasm. After centrifugation, the supernatant was collected and incubated at 37 °C with 220 r.p.m. shaking for 60 min to digest remaining mRNA with endogenous nucleases[8]. Subsequently, the extract was re-centrifuged at 12,000 × g at 4 °C for 30 min and the supernatant was transferred to 12–14 kDa molecular weight cut-off (MWCO) dialysis tubing (Spectrum Labs; Spectra/Por4) and dialyzed against 2 L of S30B buffer (14 mM Mg-glutamate, 60 mM K-glutamate, ~5 mM Tris pH 8.2) overnight at 4 °C. The following day, the extract was re-centrifuged at 12,000 × g at 4 °C for 30 min. The supernatant was optionally concentrated using a 10,000 MWCO centrifuge column (GE Healthcare; Vivaspin20) based on total protein levels from a Bradford assay (ThermoScientific) to obtain concentrations above 15 mg/mL, aliquoted, and flash frozen in liquid nitrogen before storage at −80 °C.

**Cell-free sensor optimization reactions**. Cell-free reactions were prepared by mixing 33.3% cell extract, 41.7% buffer, and 25% plasmid DNA, any inducer, and water. Buffer composition was made such that final reaction concentrations were as follows: 1.5 mM each amino acid except leucine, 1.25 mM leucine, 50 mM HEPES, 1.5 mM ATP and GTP, 0.9 mM CTP and UTP, 0.2 mg/mL tRNA, 0.26 mM CoA, 0.33 mM NAD, 0.75 mM cAMP, 0.068 mM folinic acid, 1 mM spermidine, 30 mM 3-PGA, and 2% PEG-8000. In addition, the Mg-glutamate (0–6 mM), K-glutamate (20–140 mM), and dithiothreitol (0–3 mM) levels were serially calibrated for each batch of cell extract for maximum signal. Benzoic acid, hippuric acid, and cocaine hydrochloride were purchased from Sigma-Aldrich. Permission to purchase cocaine hydrochloride was given by the French drug regulatory agency (Agence Nationale de Sécurité du Médicament et des Produits de Santé) to allow development of a new biosensor. Inducers were dissolved in ethanol and final reactions contained 0.5% ethanol for all inducer concentrations including the negative control. Reactions were prepared in PCR tubes on ice and 20 μL were transferred to a black, clear-bottom 384-well plate (ThermoScientific), sealed, and the reaction was carried out in a plate reader (Biotek; Cytation3 or Synergy HTX) to measure both endpoints and reaction kinetics. The subsequent data were processed and graphs created using custom Python scripts or Microsoft Excel. Reactions for the representative images in Fig. 2c and Fig. 3b were incubated in PCR tubes at 37 °C for 4 h and imaged on a UV table with either a Sony α6000 camera (benzoic and hippuric acid sensors) or a cell phone camera (cocaine sensor) and background subtracted with Adobe Photoshop.

**Cell-free reactions with commercial beverages or human urine**. Cell extract and buffer conditions were maintained from those used in optimization reactions. For the benzoic acid beverage sensor, 10% reaction volume of either 1× or 0.1× (diluted in water) of each beverage was added, in addition to 30 nM pBEAST-BenR and 100 nM pBen-sfGFP plasmids to 20 μL reactions containing extract and buffer. Fold change was calculated relative to a negative control reaction with water as the inducer. All beverages were purchased at a local supermarket. For the hippuric acid urine sensor, each reaction contained 10% volume of 0.1× urine, pre-diluted in water. Human urine samples were obtained from the Endocrinology Department at the University of Montpellier in accordance with ethics committee approval

(#190102). In addition, each reaction was supplemented with 0.8 U/μL of murine Rnase Inhibitor (New England Biolabs).

**Benzoic acid and hippuric acid quantification**. In order to quantify fluorescent outputs from our cell-free benzoic and hippuric acid biosensors in complex samples as a measurement of concentration, we created calibration curves by adding a range between 0 μM and 1000 μM of inducer concentrations to 20 μL cell-free reactions. Hippuric acid reactions were supplemented with 0.8 U/μL RNase inhibitor to match reaction conditions. The subsequent calibration curves were background subtracted and fit to a Hill plot in Python using: $y = (y_{max} \times x^n)/(K_D^n + x^n)$, where $y$ is the fluorescence intensity, $x$ is the inducer concentration, $y_{max}$ is the maximum fluorescence intensity, $K_D$ is the concentration of ligand needed for half-maximum binding occupation at equilibrium, and $n$ is the Hill slope. Commercial beverage benzoic acid and urine hippuric acid concentrations were then calculated by using the background-subtracted fluorescent values from those experiments as $y$ and solved for the inducer concentration $x$. Undiluted concentrations were increased by a factor of 100 to account for the 1:10 sample dilution and 10% reaction volume contribution (i.e., 2 μL sample in a 20 μL total reaction volume).

**Chemical analysis of beverage and urine by LC-MS**. The following procedure was developed for detection of benzoic and hippuric acid by ultra-high-performance LC-tandem MS. The analysis was carried out using an LCMS-8050 mass spectrometer (Shimadzu, Japan) coupled to a NexeraX2 UHPLC chain (Shimadzu, Japan). The column is a Nucleodur pyramid (1.8 μm, 50 × 2.0 mm, Macherey-Nagel) maintained at 40 °C. The eluents used were as follows: H₂O with 0.1% formic acid (A), acetonitrile with 0.1% formic acid (B). The flow rate was set to 0.5 mL/min. The injection volume was 5 μl and all the analytes were eluted over a 5 min binary gradient with a starting composition percentage of 100/0 (A/B). The LCMS-8050 is a three-quadrupole mass spectrometer with a heated electrospray ionization source. The analytes were detected in negative MRM mode. The samples were diluted by 20 in water before injection. Dihydrobenzoic acid was used as an internal standard.

**Cell-free reactions detecting cocaine via luciferase output**. To test our luciferase-output cocaine biosensor, 20 μL cell-free reactions containing CocE, TF, and reporter plasmid concentrations, 0.8 U/μL RNase inhibitor, cocaine inducer gradient, 2 μL of undiluted human urine samples, extract, and buffer were incubated at 30 °C for 8 h. Samples were then transferred to white 96-well plates and 50 μL of Luciferase Assay Reagent (Promega) was added and mixed by manual orbital agitation. The plates were sealed and luciferase levels were measured in a plate reader 2 min after addition of the reagent. Fold change was calculated relative to the 0 μM cocaine-negative control.

**Reaction models**. Coarse-grained modeling was performed using ordinary differential equations, simulated using the R software. Briefly, the model combines Michaelis–Menten kinetics for the transducer module and resource competition for RNA polymerases and ribosomes to account for varying DNA concentration effects. Michaelis–Menten equations are used for promoter activation. Production of toxic byproducts as well as energy consumption for mRNA production were also included. Full model derivation can be found in Supplementary Notes 2 and 3.

**Chemical identifiers**. In order to allow easier parsing of our article by bioinformatics tools, we provide here the identifiers of our chemical compounds:

Benzoic acid: InChI = 1 S/C7H6O2/c8-7(9)6-4-2-1-3-5-6/h1-5H,(H,8,9)

Hippuric acid: InChI = 1 S/C9H9NO3/c11-8(12)6-10-9(13)7-4-2-1-3-5-7/h1-5H,6H2,(H,10,13)(H,11,12)

Cocaine: InChI = 1 S/C17H21NO4/c1-18-12-8-9-13(18)15(17(20)21-2)14(10-12) 22-16(19)11-6-4-3-5-7-11/h3-7,12-15 H,8-10H2,1-2H3/t12-,13 + ,14-,15 + /m0/s1.

**Reporting summary**. Further information on experimental design is available in the Nature Research Reporting Summary linked to this article.

## Data availability

The source data underlying Figs. 2b, c, 3a, b, and 4a–d, and Supplementary Fig. 2 and 4-17, and DNA sequences for all constructs are provided as a Source Data File. Plasmids used in this paper are available from Addgene (pBEAST-BenR: #114597; pBEAST-pBen-sfGFP: #114598; pBEST-HipO: #114599; pBEAST-J23101-CocE: #114600; pBEAST-sfGFP: #116916; pBEAST-J23101-sfGFP: #116917; pBEAST-pBen-Luc: #122694). All other raw data are available from the corresponding authors at reasonable request.

## Code availability

Simulation scripts are available at https://github.com/brsynth. Custom python scripts used to process data are available upon request to authors.

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

## Acknowledgements

We thank the Bonnet and Faulon lab members for fruitful discussions. We are grateful to Pau Bernado, Annika Urbanek, and Anna Morato for helpful discussions on cell-free systems and extract creation, Gottfried Otting for BL21 (DE3) Star::RF1-CBD₃ strain, and the patients for providing their samples. We thank Gilles Valette from the "Laboratoire de Mesures Physiques", the analytical facilities of the University of Montpellier, where the spectrometry analyses were performed. This work was supported by an ERC starting grant "COMPUCELL" to J.B. J.B. also acknowledges continuous support from the INSERM Atip-Avenir program and the Bettencourt-Schueller Foundation. The CBS acknowledges support from the French Infrastructure for Integrated Structural Biology (FRISBI) ANR-10-INSB-05-01. J.-L.F. acknowledges support from BBSRC/EPSRC (grant number BB/M017702/1) and from the ANR (grant number ANR-15-CE1-0008). M.K. is supported by DGA (French Ministry of Defense) and Ecole Polytechnique and A.P. is supported by a scholarship from INRA and the IDEX Paris-Saclay. All plasmids will be available from Addgene.

## Author contributions

P.L.V., A.P., J.-L.F. and J.B. designed experiments. P.L.V. and A.P. cloned constructs and performed experiments. M.K. constructed the computer model simulations. I.C., P.C., E. R. and E.V. participated in clinical sample collection and analysis. P.L.V., A.P., M.K., J-L. F. and J.B wrote the paper. All authors approved the manuscript.

## Additional information

**Competing interests:** The authors declare no competing interests.

