## [Peer Review File · Nature Communications]

Reviewers' Comments:

Reviewer #1:

Remarks to the Author:

1. What are the major claims of the paper?

Voyvodic et al characterise a cell-free system with extraordinary high dynamic and operational ranges. This is particularly impactful, when considering the mode-of-action of the biosensor as a transcriptional activator.

Also, the authors illustrate the use of metabolic transducers to circumvent the potential limitations researchers may encounter when no native biosensors for ligands of choice are available. This offers a potentially generalisable method to explore HTP for ligands without any yet-characterised biosensors available.

Finally, the authors use their RetroPath algorithm to identify humane metabolome biomarkers which potentially could be coupled to ligands for which a biosensor was available.

To this reviewer's knowledge this study is the first experimental evidence of the SensiPath concept (published by co-author Dr Faulon's group (PMID: 27106061)) working in a cell-free system.

2. Are they novel and will they be of interest to others in the community and the wider field?

The claims are extrapolating from the SensiPath concept outlined by Dr Faulon's lab in *Nucleic Acids Res* (2016) (PMID: 27106061) and the experimental validation of SensiPath from in vivo studies (Libis et al., PMID: 27028723). Though the cell-free demo is new both the concept, the biosensor, the metabolic transducers tested, and the transfer function are all described previously. Also, as the authors highlight in the Introduction (Line 42-43) several studies have implemented cell-free system for real-life diagnosis using biosensors (ref 12-13, and PMID: 30131493).

3. If the conclusions are not original, it would be helpful if you could provide relevant references.

The conclusions are very much overlapping with current standards for biosensor characterisation performed in vivo. (E.g. Libis et al., PMID: 27028723; Nielsen et al, PMID: 27034378, many others).

4. Is the work convincing, and if not, what further evidence would be required to strengthen the conclusions?

Yes, the work is convincing and conclusions aligned with experimental evidence. Still, the system makes use of a biosensor and two transducers that are already well-characterized (PMID: 27028723). In reality, the authors are doing a similar characterisation of the BenR biosensor in a cell-free system as the group of Dr. Faulon did a few years ago in vivo (PMID: 27028723).

Also, the authors do not demonstrate a true application, which is presumed to be the strongest benefit of cell-free systems compared to in vivo biosensing. Would the cell-free system be able to monitor in-soil benzoate concentrations or even assist in assessing cocaine-contaminated cloths/serum? Such, real-life application would be expected to be explored if this study is to meet the general audience of *Nature Comm*.

5. On a more subjective note, do you feel that the paper will influence thinking in the field?

The authors claim that they provide "a rigorous scheme to optimise cell-free biosensor performance" (Line 197-198). This reviewer finds that the characterisation performed is largely

identical to any standard biosensor transfer function characterisation covered in detail by many other *in vivo* biosensor studies (see examples above).

Also, the SensiPath and RetroPath concepts have already been published.

Summing up; without a real-life application demonstrated, this study does not convey a lot of novelty compared to the excellent work already published by co-author Dr. Faulon. Also, unfortunately, this submission seems to miss Supplementary table 3-4. A shame, as those findings would be very interesting to see.

6. Recommended changes:

Majors:

Supplementary tables 3 and 4 are missing.

Minors:

Line 107-108: Plasmid conc not Plasmids conc.

Line 141-142: What is meant by sensitivity? Is this referring to cooperative action between BenR monomers or allosteric cooperativity? Or is this referring to the affinity of BenR to benzoic acid. I recommend to use the term sensitivity for the steepness of slope of the transfer function. In fact, as judged from the the slope of the transfer function in Libis et al, this reviewer do not agree that the cell-free biosensor offer higher sensitivity to benzoate (slope is steeper *in vivo*) compared to *in vivo* system. Indeed, operational range is shifted towards lower detection limits of benzoate.

Line 190: Response in stead of responses

Line 204: remove "that"

Reviewer #2:

Remarks to the Author:

The manuscript by Voyvodic et al. describes a novel way to increase the sensing capabilities of cell-free systems. Rather than engineering new sensors, the authors exploit enzymes that convert chemicals that cannot be normally sensed into molecules for which a ligand responsive transcriptional activator already exists. The goal is clearly stated in the manuscript, and the data clearly support the conclusions. The manuscript is a bit short, but a longer manuscript would, in my opinion, detract from the message.

More specifically, the authors exploit a sensor for benzoic acid (Figure 2 and Supplementary Figure 1) using the cognate receptor BenR and the expression of super folder GFP. Then to expand the detection capabilities of the system, an algorithm (RetroPath) in conjunction with the SensiPath webserver was used to identify 64 molecules that could be transformed into benzoic acid in one enzymatic step. Two of these molecules (hippuric acid and cocaine) were chosen for testing. The ability to detect hippuric acid and cocaine relied upon three different plasmids. One plasmid coded for an enzyme to transform the analyte into benzoic acid (either HipO or CocE). Another plasmid coded for the transcription factor BenR, and the last plasmid coded for the sfGFP reporter gene. The first two plasmids were under the control of a constitutive promoter while the reporter was under the control of an inducible pBen promoter. The endogenous promoters, however, were different (OR2-OR1-Pr for the HipO hippurase and J23101 for the CocE esterase). It is unclear if

there was a reason for the different constitutive promoters. Supplemental figure 3 seems to show that the use of J23101 gives less protein, which explains why the optimal concentration of plasmid for CocE was higher than for HipO.

In my opinion, the described strategy has numerous applications in sensing technologies for both cell-free and in vivo systems and could be exploited for the wiring of more complex pathways. Therefore, I expect the authors and others to build upon the work in several different interesting and useful directions.

From what is described in the manuscript, it is fairly straightforward to differentiate from benzoic acid and either hippuric acid and cocaine, but that is not explicitly explained. I'd suggest spelling this out in the manuscript. Similarly, could simple logic gates be constructed to help differentiate analytes?

Reproducibility would likely be improved, and perhaps facilitate the modeling as well, if all of the necessary genes were placed on the same plasmid as opposed to using three different plasmids.

Although I find the presented work novel, there are some conceptual similarities to previous work on artificial cells (DOI: 10.1038/ncomms5012).

In summary, I believe this work helps advance the field in a way that has not been articulated previously and thus benefits efforts in cell-free and in vivo synthetic biology and biotechnology.

Reviewer #3:

Remarks to the Author:

The work by Voyvodic et al claims to provide a framework to expand the range of molecules detectable by cell-free biosensors by combining synthetic metabolic cascades with sensors using known or well-characterized transcription factors. They present biosensors with response on the order of hours and a high dynamic range.

The existence of a new framework for expanding the space of metabolic sensors would indeed be impactful, as our approaches to developing new sensors are largely limited. Many current approaches rely on bioprospecting or database mining to find existing macromolecular sensors of molecules that can be dropped into cell-free or whole-cell biosensors. Another common alternative is the effort to design riboswitches, though these are notoriously hard to make and have not yet been generalized. Another alternative that has been used in the past is to harness the activity of an enzyme that is capable of producing some output that is in fact detectable. Lots of electrochemical sensors work this way. Perhaps glucose oxidase is one of the most well-known: glucose is not directly measured, but rather the activity of an enzyme on glucose is captured by measuring redox of the byproduct. There are numerous other electrochemical examples, including those that measure NADPH with either chromogenic or fluorescent readouts. Another example of this strategy (using chemical reaction to make something more easily measurable) can be seen in the development of higher-throughput screens in metabolic engineering, often using things like visible pigments. See, for example, Santos and Stephanopoulos in 2008 using tyrosine conversion to melanin to enable their metabolic engineering efforts.

As stated at the beginning of the last paragraph, a new framework for expanding the space of metabolic sensors would indeed be impactful. However, as is evident from the rest of that paragraph, the strategy presented by the authors is not really new or novel, either in terms of biosensor development or applications of those biosensor. This could be the first time that such an approach has been used in cell-free systems. However, to this reviewer there is nothing inherently novel or unique about what the authors are proposing, at least on the scale that is presented. If the authors had shown that they could quickly generate many new sensors with many different

sensor modules plugged into it, that could have been more persuasive as to the generalizability and true "framework" nature and impact of the work presented.

Rather, the authors take a sensor module previously characterized *in vivo* and first show that it works in cell-free systems. They then use other previously-published software to select a metabolic enzyme that can catalyze the conversion of some other molecule into their target molecule, they add that enzyme to their reactions, show that those sensors work, and then suggest the overarching success of their framework. So really, the true novelty of the work presented here is that there are now two *in vitro* sensors, one for cocaine and one for hippuric acid. The novel framework argument is hard to interpret as being as impactful as the authors claim, the novelty lies on the application side.

On the impact of those two sensors, the authors also fall short. They test their measurements in synthetic mixtures. Is that the environment the sensors would really be used in? Is that the appropriate sample matrix? Can their cell-free system even function in their target matrix or matrices? For example, work from the Bundy group has shown that significant efforts may need to be taken to get systems to work in complex biofluid matrices, though it may ultimately be possible. But the authors do not present any real application such as this, only a laboratory proof-of-principle for two biomarkers, one of which (hippuric acid) is not really well-justified for why the reader should be so interested in it.

Another major contribution that the authors claim is their mathematical model. While the main text contains very little mention of the model, the supplement begins to expand, and then the appendix further expands. There are some uncertainties as to decisions that are being made, etc., but one can readily ask, without assessment of the model's validity/veracity: what contribution is it making to the paper? How does the existence of the model help the authors prove or do anything that the experiments do not do? And has the predictivity of the model been validated in any way? Unfortunately, the answers to most of these questions are not positive for the authors.

It is unclear how the model contributes any new knowledge. The authors conclude that resource limitations ultimately may have caused DNA concentration sensitivity. That is probably one of the first hypotheses any cell free scientist would have had before a model. And the authors have not taken any efforts to try to validate that hypothesis. They have multiple dimensions of DNA concentrations they can vary, they have left two fixed and not considered how those could test their hypothesis. They also have not attempted to characterize anything about their system with their model. There does not appear to be sensitivity analysis to gain a deeper understanding of what it means or to see what can be learned. Moreover, the authors ultimately say in the supplement that they really only tried to make a qualitative model, and that quantitative models are unrealistic bordering on pointless due to variations in experiments. They even allude to the fact that their model may not actually capture the biological effects underpinning the data. Taken together, then, while the formal definition of the model in mathematical language is extensive, the justification for and impact of the model does not appear to be substantial.

Taken together, then, there are some concerns about the novelty and impact of this work. However, please do not get this reviewer wrong: the work that is presented is readily publishable in any of a number of journals. There do not appear to be major methodological flaws. However, the scope of work performed and the impact leave something to be desired, especially for the journal to which the manuscript has been submitted.

In addition to these overarching comments, here are additional itemized, roughly chronological comments/suggestions/minor concerns:

- Overall: the idea that direct detection for non-benzoic acid sensors would be easy rests on the assumption that all of the responses from other transcription factors would be similar. What if some are more switch-like and some are more linear? Could they still get the same quantitative responses? This concern undercuts the potential generalizability without demonstration of

something to that effect.

- Overall: the authors talk about biomarkers being important, which sounds compelling, but looking at their biomarker list, some are entire panels for one disease (20 or more biomarkers) which would decrease the importance of individual sensors, and other molecules can be biomarkers for 2, 5, or even 20+ diseases, which affects the diagnostic importance of the information they would provide.

- Overall: why did the authors use a different promoter, different plate reader, different camera, etc. for one set of experiments? Is this providing benefit? Or is it perhaps just adding potential confounders?

- Page 4: why did the authors not push the reporter plasmid concentrations higher? They are clearly not at a local optimum yet, though they may be for the sensor module. Could that perhaps push sensitivity higher or improve other sensor parameters? Would it change whether or not the sensor module has a plateau point, or where it is?

- Page 5: The authors allude to greater sensitivity than previous work but provide no quantitative comparison.

- Page 5: The authors' language here is too strong. They claim to "conclude" and "identify" and prove things, but given their scope of work they have not done that. They have only generated some modest hypotheses, and not tested any of them.

- Page 7 first paragraph, the authors seem to suggest that the low cross-reactivity will be generalizable. This is very likely specific to their work. Their two targets are very, very far away in chemical space from benzoic acid. The only thing they have in common is the aromatic ring. The targets have huge functional groups that could obviously be the cause of that specificity for the transcription factor, but other targets may not.

- The authors appear to have changed scales for their graphs? This greatly, greatly complicates interpretation of the data. Please keep them constant, especially when claims are made (as the authors do) about the quantitative results being the same across the different sensors.

- Page 7, the authors claim to provide a "rigorous scheme to optimize cell-free biosensor performance"; no such novel rigorous scheme is provided. They just tested a bunch of concentrations, as any engineer might have done.

- Page 7, the authors claim the model will help future development. How? See above analyses.

- Page 7, the authors say they have 64 biomarkers but the supplement says general metabolites. The supplementary file includes molecules like carbon dioxide and hydroxide, suggesting it is not just biomarkers.

- Page 7, additional context should be given to those biomarkers. If a disease diagnosis requires 20 markers to be measured, is this much different than running an existing clinical panel or using analytical chemistry techniques like mass spectrometry?

- Page 11: Why did they "change" the promoter for CocE? Justification for an arbitrary decision like that is needed, especially when the authors then later blame this decision in part for problems in their model development.

- Is a Hill equation really well-justified? Where would the cooperativity come in? This assumption should be justified better before adding another adjustable parameter.

- The authors appear to have a typo with different terms K_{tx} and K_{tox} .

- The meaning of the gamma, pi, lambda, and delta terms should be made more evident even in the absence of the detailed appendix. This needs to be digestible, especially when the authors claim that nothing needed to be fitted and all of the data came from the literature.

- Page 20, the authors say they made the affinity "smaller". Do they mean "weaker"? Because that would be quantitatively "bigger", not "smaller". ("greater", not "lesser", magnitude)

- Page 20, shouldn't the impact of using a different plate reader be taken away by the normalization of fluorescent signals to a fixed concentration of strong promoter-driven GFP?

- Page 21: The authors identified a shortcoming in their model (spontaneous transformation), and a potential solution and didn't bother to fix it? The justification here seems weak.

- Why do both hippuric acid and cocaine have a larger baseline measurement with no enzyme at 500, 1000 μ M than benzoic acid?

- The authors seem to have gotten some parameter values from established databases of in vitro or in vivo (but not cell free) parameters, and others from published papers of in vivo work. Is

there really a good reason to expect those to all just work perfectly like for hippuric acid? Or is there perhaps not much sensitivity once they fix a few parameters to their system?

- Why did the authors divide rate parameters by 10? Though the step was well-disclosed, this reviewer finds some difficulty in finding the explicit justification.

Response to reviewers: “Plug-and-Play Metabolic Transducers Expand the Chemical Detection Space of Cell-Free Biosensors”

Peter L Voyvodic, Amir Pandi, Mathilde Koch, Ismael Conejero, Emmanuel Valjent, Philippe Courtet, Eric Renard, Jean-Loup Faulon*†, and Jerome Bonnet*†

5

We would like to thank the reviewers for their constructive comments which we believe have improved the quality of the paper. Below is a full point-by-point response to the reviewer's comments. Our answers are in blue.

10 **Reviewers' comments:**

Reviewer #1 (Remarks to the Author):

1. What are the major claims of the paper?

15 Voyvodic et al characterise a cell-free system with extraordinary high dynamic and operational ranges. This is particularly impactful, when considering the mode-of-action of the biosensor an transcriptional activator.

Also, the authors illustrate the use of metabolic transducers to circumvent the potential limitations researchers may encounter when no native biosensors for ligands of choice are available. This offers a potentially generalisable method to explore HTP for ligands without any yet-characterised biosensors available.

20 Finally, the authors use their RetroPath algorithm to identify humane metabolome biomarkers which potentially could be coupled to ligands for which a biosensor was available.

To this reviewer's knowledge this study is the first experimental evidence of the SensiPath concept (published by co-author Dr Faulon's group (PMID: 27106061)) working in a cell-free system.

25 2. Are they novel and will they be of interest to others in the community and the wider field?

30 The claims are extrapolating from the SensiPath concept outlined by Dr Faulon's lab in Nucleic Acids Res (2016)(PMID: 27106061) and the experimental validation of SensiPath from in vivo studies (Libis et al., PMID: 27028723). Though the cell-free demo is new both the concept, the biosensor, the metabolic transducers tested, and the transfer function are all described previously. Also, as the authors highlight in the Introduction (Line 42-43) several studies have implemented cell-free system for real-life diagnosis using biosensors (ref 12-13, and PMID:

30131493).

35 **Answer:** Inspired by the reviewer's suggestions to implement the cell-free system for real-world applications, we have created applications to detect benzoic acid in products from the food industry and to detect hippuric acid and cocaine in human urine (**Figure 4**). See later response for more details.

3. If the conclusions are not original, it would be helpful if you could provide relevant references.

40 The conclusions are very much overlapping with current standards for biosensor characterisation performed in vivo. (E.g. Libis et al., PMID: 27028723; Nielsen et al, PMID: 27034378, many others).

Answer: See previous answer.

4. Is the work convincing, and if not, what further evidence would be required to strengthen the conclusions?

45 Yes, the work is convincing and conclusions aligned with experimental evidence. Still, the system makes use of a biosensor and two transducers that are already well-characterized (PMID: 27028723). In reality, the authors are doing a similar characterisation of the BenR biosensor in a cell-free system as the group of Dr. Faulon did a few years ago in vivo (PMID: 27028723).

50 Also, the authors do not demonstrate a true application, which is presumed to be the strongest benefit of cell-free systems compared to in vivo biosensing. Would the cell-free system be able to monitor in-soil benzoate concentrations or even assist in assessing cocaine-contaminated clothings/serum? Such, real-life application would be expected to be explored if this study is to meet the general audience of Nature Comm.

55 **Answer:** We thank the reviewer for his/her suggestions for real-world applications. After conducting a literature search for potential uses of diagnostics for benzoic acid, hippuric acid, and cocaine, we settled on three applications that would be useful and of interest to the general audience of **Nature Communications**.

60 Benzoic acid and sodium benzoate are common food additives in products ranging from energy drinks to marmalades. They are classified by the United States Food and Drug Administration as Generally Recognized as Safe (GRAS); however, the maximum allowable levels in foods and beverages are regulated and some patients with chronic inflammation or orofacial granulomatosis exhibit adverse reactions and are placed on benzoate-free diets by their physicians. As a proof-of-concept of our benzoic acid sensor's functionality in response to
65 a complex inducer, we directly added commercial beverages and demonstrated that our

cell-free benzoic acid sensor is capable to detecting benzoates in a range of beverages in under an hour with strong signal strength (fold change up to ~180) (**Fig. 4a**). Additionally, we obtained quantitative benzoic acid results using 10-fold diluted beverage and compared them to measurements by liquid chromatography-mass spectrometry (LC-MS) (**Fig. 4b**).

Hippuric acid is common metabolite found in urine, the result of liver metabolism of a range of aromatic compounds. While previous studies have found correlation between higher levels of hippuric acid and exposure to toluene in occupational conditions (DOI: 10.1136/oem.35.4.330), we were inspired by recent work by **Isabella et al.** (DOI: 10.1038/nbt.4222) using hippuric acid as a biomarker for the efficacy of an orally-consumed modified strain of **E. coli** that converts phenylalanine into **trans**-cinnamate in patients suffering from phenylketonuria. In testing human urine with our cell-free sensor, we were able to detect and quantify the levels of hippuric acid in urine (**Fig. 4c**). Additionally, we confirmed our results using LC-MS, providing a proof-of-concept example of a cell-free hippuric acid diagnostic that could be used in conjunction with other upcoming synthetic biology-based medical treatments.

Finally, shortly after consumption of cocaine, levels in the urine have been detected upwards of 100 μM (DOI: 10.1093/jat/24.7.478). To see if our system was capable of detecting these clinically-relevant levels, we spiked a titration of cocaine levels into human urine to see lower limit that we were able to detect. Our experiment showed successful detection with an 8-fold change at 10 μM and at least 2-fold change at values as low as 3 μM . We believe these proof-of-concept examples illustrate the future ability of cell-free biosensors to be used by the food and beverage industry, medical professionals, and drug enforcement agencies.

5. On a more subjective note, do you feel that the paper will influence thinking in the field?

The authors claim that they provide “a rigorous scheme to optimise cell-free biosensor performance” (Line 197-198). This reviewer finds that the characterisation performed is largely identical to any standard biosensor transfer function characterisation covered in detail by many other in vivo biosensor studies (see examples above).

Answer: We thank the reviewer for his/her suggestion. Part of the novelty of our cell-free system optimization procedure was the ability to individually titrate each DNA component to maximize signal fold change and dynamic range. To highlight this aspect, the above sentence has been changed to: “In addition, we provide a three order-of-magnitude titration for each DNA component to optimize cell-free biosensor performance along with a mathematical model enabling a better understanding of the parameters governing cell-free biosensors response which will help future optimisation of such devices.”

Also, the SensiPath and RetroPath concepts have already been published.

Summing up; without a real-life application demonstrated, this study does not convey a lot of novelty compared to the excellent work already published by co-author Dr. Faulon. Also,

unfortunately, this submission seems to miss Supplementary table 3-4. A shame, as those findings would be very interesting to see.

Answer: We thank the reviewer for all of his/her thoughtful suggestions that clearly have improved the paper. We targeted three real-life applications, and demonstrated that each of our sensors can function in complex samples to detect benzoic acid in commercial beverages, endogenous hippuric acid in urine, and spiked cocaine in urines at clinically relevant concentrations. We believe that these additions will be a great interest to the **Nature Communications** readership community. Additionally, we regret if Supplementary Tables 3 and 4 (now Supplementary Tables 6 and 7) were not found in the submission. As they are quite large in size, we included them as their own files. We hope that the reviewer can now access them in this resubmission.

6. Recommended changes:

Majors:

Supplementary tables 3 and 4 are missing.

Answer: As stated in the previous answer, Supplemental Tables 3 and 4 (now Supplementary Tables 6 and 7) can be found as Supplementary Excel Files in this resubmission.

Minors:

Line 107-108: Plasmid conc not Plasmids conc.

Answer: The typo has been corrected.

Line 141-142: What is meant by sensitivity? Is this referring to cooperate action between BenR monomers or allosteric cooperativity? Or is this referring to the affinity of BenR to benzoic acid. I recommend to use the term sensitivity for the steepness of slope of the transfer function. In fact, as judged from the the slope of the transfer function in Libis et al, this reviewer do not agree that the cell-free biosensor offer higher sensitivity to benzoate (slope is steeper in vivo) compared to in vivo system. Indeed, operational range is shifted towards lower detection limits of benzoate.

Answer: We thank the reviewer for bringing up this point. We agree that the operational range is shifted towards lower detection limits. The text has been changed to emphasize this point and avoid confusion. However, according to the definition of sensitivity, we also want to emphasize that our system is indeed more sensitive than the **in vivo** version, by at least an order of magnitude (~ 0.1 fold-change/ μM -inducer **in vivo** between benzoic acid concentrations of 25 μM and 100 μM vs. 1.2 fold-change/ μM -inducer in our cell-free system between benzoic

acid concentrations of 20 μM and 100 μM). In fact, if one looks at Libis et. al, 2016, Figure 4C which represents fold change vs. inducer concentration, like we do in Figures 2C and 3B, the higher sensitivity of our system is clearly visible, although, a confusion might arise due to the fact that **in vivo** fold changes are represented in linear scale while cell-free fold changes are in log scale due to the higher maximum value.

Line 190: Response instead of responses

Answer: The typo has been corrected.

Line 204: remove “that”

Answer: The typo has been corrected.

Reviewer #2 (Remarks to the Author):

The manuscript by Voyvodic et al. describes a novel way to increase the sensing capabilities of cell-free systems. Rather than engineering new sensors, the authors exploit enzymes that convert chemicals that cannot be normally sensed into molecules for which a ligand responsive transcriptional activator already exists. The goal is clearly stated in the manuscript, and the data clearly support the conclusions. The manuscript is a bit short, but a longer manuscript would, in my opinion, detract from the message.

More specifically, the authors exploit a sensor for benzoic acid (Figure 2 and Supplementary Figure 1) using the cognate receptor BenR and the expression of super folder GFP. Then to expand the detection capabilities of the system, an algorithm (RetroPath) in conjunction with the SensiPath webserver was used to identify 64 molecules that could be transformed into benzoic acid in one enzymatic step. Two of these molecules (hippuric acid and cocaine) were chosen for testing. The ability to detect hippuric acid and cocaine relied upon three different plasmids. One plasmid coded for an enzyme to transform the analyte into benzoic acid (either HipO or CocE). Another plasmid coded for the transcription factor BenR, and the last plasmid coded for the sfGFP reporter gene. The first two plasmids were under the control of a constitutive promoter while the reporter was under the control of an inducible pBen promoter. The endogenous promoters, however, were different (OR2-OR1-Pr for the HipO hippurase and J23101 for the CocE esterase). It is unclear if there was a reason for the different constitutive promoters. Supplemental figure 3 seems to show that the use of J23101 gives less protein, which explains why the optimal concentration of plasmid for CocE was higher than for HipO.

Answer: During cloning we found that using the same Pr-OR2-OR1 promoter as the BenR and HipO plasmids for CocE led to lethal toxicity or mutation. We had previously tried using the *E. coli* strain KL740 (Coli Genetic Stock Center (CGSC)# 4382) with a temperature-sensitive lambda phage repressor, but had been able to obtain successful function. Thus, in an effort to create a successful clone to grow our plasmid, we switched to another canonical strong constitutive promoter, J23101 from the Anderson collection. While the strengths of the promoters are not identical, we were impressed with our system's ability to compensate for promoter strength during our calibration process.

In my opinion, the described strategy has numerous applications in sensing technologies for both cell-free and in vivo systems and could be exploited for the wiring of more complex pathways. Therefore, I expect the authors and others to build upon the work in several different interesting and useful directions.

From what is described in the manuscript, it is fairly straightforward to differentiate from benzoic acid and either hippuric acid and cocaine, but that is not explicitly explained. I'd suggest spelling this out in the manuscript. Similarly, could simple logic gates be constructed to help differentiate analytes?

Answer: We thank the reviewer for this suggestion and have added a sentence to spell that out. Regarding logic gates, it is true that we could engineer logic gates to differentiate the different molecules in the same sample. One option would be to combine the current circuits with new ones that would detect the by-products of metabolic reactions which are different in each case (e.g. for HipO and CocE). This is a possibility, however the convenience and ease-of-use of the cell-free systems allows us to perform the different reactions in parallel, with much simpler designs.

It is however unclear why one would like to build a sensing system that differentiates cocaine from hippurate. Yet, if such a device was to be build, rather than using a logic gate, another option would be to use different effectors/transcription factors for the two compounds. For instance cocaine could be detected by the CocE/Benzoate/BenR systems we presented in the manuscript while hippurate could be detected by another system (for instance Glyat (EC 2.3.1.71)/Benzoyl-CoA/BadR [PMID:10094687])

Reproducibility would likely be improved, and perhaps facilitate the modeling as well, if all of the necessary genes were placed on the same plasmid as opposed to using three different plasmids.

Answer: While the experimental setup and modeling would indeed be simplified by placing all of the necessary genes on the same plasmid, unfortunately that would eliminate the possibility of independently titrating the DNA concentration of our components. This is a critical component to our sensor optimization process and, we believe, an important part of this work. Indeed, when we began this project we used the transcription factor and reporter on the same plasmid, as had been previously done in vivo, but found that our sensors were much more tunable with components on individual plasmids. We have slightly modified the text to clarify the interest of this method.

Although I find the presented work novel, there are some conceptual similarities to previous work on artificial cells (DOI: 10.1038/ncomms5012).

Answer:

We thank the reviewer for the suggestion and have added this reference to our introduction. Indeed this paper, like ours, builds on the general principle of using an intermediate device to convert a signal normally undetectable by the cells into a detectable one, using cell-free encapsulated communicating with living cells.

One major difference we believe, beyond the physical implementation of the two systems, is that our system using metabolic transducers can provide a more quantitative response, as seen in Figure 4.

In summary, I believe this work helps advance the field in a way that has not been articulated previously and thus benefits efforts in cell-free and in vivo synthetic biology and biotechnology.

We thank the reviewer for her/his comments that helped clarify some key points of the paper.

Reviewer #3 (Remarks to the Author):

The work by Voyvodic et al claims to provide a framework to expand the range of molecules detectable by cell-free biosensors by combining synthetic metabolic cascades with sensors using known or well-characterized transcription factors. They present biosensors with response on the order of hours and a high dynamic range.

The existence of a new framework for expanding the space of metabolic sensors would indeed be impactful, as our approaches to developing new sensors are largely limited. Many current approaches rely on bioprospecting or database mining to find existing macromolecular sensors of molecules that can be dropped into cell-free or whole-cell biosensors. Another common alternative is the effort to design riboswitches, though these are notoriously hard to make and have not yet been generalized. Another alternative that has been used in the past is to harness the activity of an enzyme that is capable of producing some output that is in fact detectable. Lots of electrochemical sensors work this way. Perhaps glucose oxidase is one of the most well-known: glucose is not directly measured, but rather the activity of an enzyme on glucose is captured by measuring redox of the byproduct. There are numerous other electrochemical examples, including those that measure NADPH with either chromogenic or fluorescent readouts. Another example of this strategy (using chemical reaction to make something more easily measurable) can be seen in the development of higher-throughput screens in metabolic engineering, often using things like visible pigments. See, for example, Santos and Stephanopoulos in 2008 using tyrosine conversion to melanin to enable their metabolic engineering efforts.

As stated at the beginning of the last paragraph, a new framework for expanding the space of metabolic sensors would indeed be impactful. However, as is evident from the rest of that paragraph, the strategy presented by the authors is not really new or novel, either in terms of biosensor development or applications of those biosensor. This could be the first time that such an approach has been used in cell-free systems. However, to this reviewer there is nothing inherently novel or unique about what the authors are proposing, at least on the scale that is presented. If the authors had shown that they could quickly generate many new sensors with many different sensor modules plugged into it, that could have been more persuasive as to the generalizability and true "framework" nature and impact of the work presented.

Answer: We thank the reviewer for his/her thorough analysis of our manuscript and acknowledge his/her in helping strengthen this paper. We respectfully disagree with reviewer #3 regarding the novelty of the work, but we did our best to address all the comments in order to

convince him/her. In particular, we provide three meaningful applications as described below.

Rather, the authors take a sensor module previously characterized in vivo and first show that it works in cell-free systems. They then use other previously-published software to select a metabolic enzyme that can catalyze the conversion of some other molecule into their target molecule, they add that enzyme to their reactions, show that those sensors work, and then suggest the overarching success of their framework. So really, the true novelty of the work presented here is that there are now two in vitro sensors, one for cocaine and one for hippuric acid. The novel framework argument is hard to interpret as being as impactful as the authors claim, the novelty lies on the application side.

Answer: We indeed provide new applications (see below). We changed the word “framework” which might have created confusion to “workflow”. It is correct that we use sensipath to determine metabolic transducers and transcription factors that were previously characterized in vivo. However, from there the implementation process differs significantly from the one for in vivo circuits. Original contributions from our work include: 1) modularization of circuit components on different vectors; 2) combinatorial titration to identify optimal working concentrations; 3) optimization of our system in the context of resource depletion occurring in cell-free systems; 4) new applications for cell-free biosensors in the context of complex samples including beverages and clinical samples, without any pre-processing. In particular, the detection of endogenous biomarkers in pure samples. Point 1 to 3 have never been presented to that degree of details and we make here an important contribution to the field of cell-free biosensor engineering on which other groups can build. Point 4: this is actually the first time a endogenous biomarker is detected at clinically relevant concentrations without sample preprocessing. More details below.

On the impact of those two sensors, the authors also fall short. They test their measurements in synthetic mixtures. Is that the environment the sensors would really be used in? Is that the appropriate sample matrix? Can their cell-free system even function in their target matrix or matrices? For example, work from the Bundy group has shown that significant efforts may need to be taken to get systems to work in complex biofluid matrices, though it may ultimately be possible. But the authors do not present any real application such as this, only a laboratory proof-of-principle for two biomarkers, one of which (hippuric acid) is not really well-justified for why the reader should be so interested in it.

Answer: While we believe that our sensors showed impressive fold change and dynamic range, we too acknowledge that the addition of synthetic mixtures is not the end-goal of our cell-free biosensors. To this end, we show that our sensors can operate a complex environment that can have significant matrix effects: commercial beverages and clinical samples (urine). We have found that the sensors operated relatively well in complex media, and indeed were able to improve the lower detection limit of the cocaine sensor by using the luciferase reporter.

Benzoic acid and sodium benzoate are common food additives in products ranging from energy drinks to marmalades. They are classified by the United States Food and Drug Administration as Generally Recognized as Safe (GRAS); however, the maximum allowable levels in foods and beverages are regulated and some patients with chronic inflammation or orofacial granulomatosis exhibit adverse reactions and are placed on benzoate-free diets by their physicians. As a proof-of-concept of our benzoic acid sensor's functionality in response to a complex inducer, we directly added commercial beverages and demonstrated that our cell-free benzoic acid sensor is capable to detecting benzoates in a range of beverages in under an hour with strong signal strength (fold change up to ~180) (**Fig. 4a**). Additionally, we obtained quantitative benzoic acid results using 10-fold diluted beverage and compared them to measurements by liquid chromatography-mass spectrometry (LC-MS) (**Fig. 4b**).

Hippuric acid is common metabolite found in urine, the result of liver metabolism of a range of aromatic compounds. While previous studies have found correlation between higher levels of hippuric acid and exposure to toluene in occupational conditions (DOI: 10.1136/oem.35.4.330), we were inspired by recent work by **Isabella et al.** (DOI: 10.1038/nbt.4222) using hippuric acid as a biomarker for the efficacy of an orally-consumed modified strain of **E. coli** that converts phenylalanine into **trans**-cinnamate in patients suffering from phenylketonuria. In testing human urine with our cell-free sensor, we were able to detect and quantify the levels of hippuric acid in urine (**Fig. 4c**). Additionally, we confirmed our results using LC-MS, providing a proof-of-concept example of a cell-free hippuric acid diagnostic that could be used in conjunction with other upcoming synthetic biology-based medical treatments.

Finally, shortly after consumption of cocaine, levels in the urine have been detected upwards of 100 μM (DOI: 10.1093/jat/24.7.478). To see if our system was capable of detecting these clinically-relevant levels, we spiked a titration of cocaine levels into human urine to see lower limit that we were able to detect. Our experiment showed successful detection with an 8-fold change at 10 μM and at least 2-fold change at values as low as 3 μM . We believe these proof-of-concept examples illustrate the future ability of cell-free biosensors to be used by the food and beverage industry, medical professionals, and drug enforcement agencies.

Another major contribution that the authors claim is their mathematical model. While the main text contains very little mention of the model, the supplement begins to expand, and then the appendix further expands. There are some uncertainties as to decisions that are being made, etc., but one can readily ask, without assessment of the model's validity/veracity: what contribution is it making to the paper? How does the existence of the model help the authors prove or do anything that the experiments do not do? And has the predictivity of the model been validated in any way? Unfortunately, the answers to most of these questions are not positive for the authors.

Answer: While we do believe that formalising knowledge and verifying coherence between expected behavior and typical biological parameters is useful for the field of synthetic biology in itself, we do understand the reviewer's concerns. To that end, we used the model to test

other concentration combinations of reporter, BenR, and enzyme DNA that have not been experimentally tested. While most of those results showed less efficient biosensors, one configuration was thought to be of interest to validate the model. As the benzoic acid calibration results were similar for 30 nM and 100 nM transcription factor DNA concentration (with 100 nM reporter DNA concentration), we were interested in using the model to see how our metabolic hybrid sensor calibration would have changed with a different transcription factor concentration. The model predicted a shift to higher enzyme DNA concentrations to obtain the same response. We then experimentally validated this shift (**Supplementary Figure 5**), supporting our hypothesis of resource competition model and increasing our confidence in the model's ability to predict sensor characteristics at different DNA concentrations. These findings were developed in a new paragraph in the **Supplementary Text** entitled 'Model prediction experimental demonstration'.

It is unclear how the model contributes any new knowledge. The authors conclude that resource limitations ultimately may have caused DNA concentration sensitivity. That is probably one of the first hypotheses any cell free scientist would have had before a model. And the authors have not taken any efforts to try to validate that hypothesis. They have multiple dimensions of DNA concentrations they can vary, they have left two fixed and not considered how those could test their hypothesis. They also have not attempted to characterize anything about their system with their model. There does not appear to be sensitivity analysis to gain a deeper understanding of what it means or to see what can be learned. Moreover, the authors ultimately say in the supplement that they really only tried to make a qualitative model, and that quantitative models are unrealistic bordering on pointless due to variations in experiments. They even allude to the fact that their model may not actually capture the biological effects underpinning the data. Taken together, then, while the formal definition of the model in mathematical language is extensive, the justification for and impact of the model does not appear to be substantial.

Answer: The first aim of model development was to help us understand why the two enzymes behaved differently and why we observed a bell-shaped response while increasing concentration of enzyme DNA. In response to the reviewer's comment, the model has been modified to include the BenR reporter array as well. Therefore, each DNA concentration can now be adjusted in the model (see above response for experimental verification of one predicted effect). While understanding the effects of parameters in the model was gained during development, this was not seen as the main focus of the paper and not insisted upon. What we meant concerning the possible lack of explanation of the biological effects underpinning the data is only about the cell-free exhaustion: the model adequately captures biological effects such as resource competition and effects of changing promoters and DNA concentrations. However, there are multiple explanations as to why cell-free systems exhaust over time and no definite experimental answers, meaning that multiple modeling strategies can reproduce the effect by using any of the current hypothesis. The paragraph discussion model limitations has been rephrased in the supplementary material.

Taken together, then, there are some concerns about the novelty and impact of this work. However, please do not get this reviewer wrong: the work that is presented is readily publishable in any of a number of journals. There do not appear to be major methodological flaws. However, the scope of work performed and the impact leave something to be desired, especially for the journal to which the manuscript has been submitted.

Answer: Again, we believe our work present sufficient novelty and impact, especially after conceptual and technical clarifications and novel real-world applications provided to be published in **Nature Communications**.

In addition to these overarching comments, here are additional itemized, roughly chronological comments/suggestions/minor concerns:

- Overall: the idea that direct detection for non-benzoic acid sensors would be easy rests on the assumption that all of the responses from other transcription factors would be similar. What if some are more switch-like and some are more linear? Could they still get the same quantitative responses? This concern undercuts the potential generalizability without demonstration of something to that effect.

Answer: The reviewer raises an interesting point. The rate of chemical reactions is not limiting **per se** from what we show here as the total speed of the reaction as well as fold changes are not significantly altered in BenR only or in HipO and CocE sensors. The main issue we agree therefore lies into the use of other transcription factors sensors that could have different behavior, in particular low fold change. This is actually an issue with all TF-based biosensors, and many methods could be applied to improve this response, in particular, the use of genetic amplifiers, as described previously by us (DOI: 10.1126/scitranslmed.aaa3601). Work around promoter/TF engineering would also be useful. Nevertheless, in this context, our work is even more relevant, because once we have a working transcription factor, we can use it to detect many different molecules by plugging various metabolic transducers, with minimal optimization required, instead of looking for and optimizing a new transcription factor.

- Overall: the authors talk about biomarkers being important, which sounds compelling, but looking at their biomarker list, some are entire panels for one disease (20 or more biomarkers) which would decrease the importance of individual sensors, and other molecules can be biomarkers for 2, 5, or even 20+ diseases, which affects the diagnostic importance of the information they would provide.

Answer: In Supplementary Table 3 (now Supplementary Table 6), we find that ~25% of biomarkers are shared by at least two diseases. Therefore, while one can develop biosensors and repurpose them for several diseases, biosensors can also be designed for a panel of biomarkers specific to a given disease. Evidently, to diagnose a disease one will need to detect more than one biomarker, this is not novel finding but already known from metabolomics studies. To answer the reviewer comment we added the following sentence in the supplementary text 'SensiPath Metabolic Space Analysis' section: "Finally, we found that

~25% biomarkers were shared by at least two diseases. Therefore, while one can develop biosensors and repurpose them for several diseases, biosensors can also be designed for a panel of biomarkers specific to a given disease”

- Overall: why did the authors use a different promoter, different plate reader, different camera, etc. for one set of experiments? Is this providing benefit? Or is it perhaps just adding potential confounders?

Answer: When we started the project, the permission to use cocaine hydrochloride by the French drug regulatory agency (Agence Nationale de Sécurité du Médicament et des Produits de Santé) was limited to the co-authors located at the Micalis Institute, while the other cell-free experiments were performed at the Centre de Biochimie Structurale. This regulatory hurdle necessitated the use of a different plate reader and camera for that experiment; however, we feel that the similar transfer function and fold changes of biosensor response between experiments strengthens the case for reproducibility.

Additionally, during cloning we found that using the same Pr-OR2-OR1 promoter as the BenR and HipO plasmids led to lethal toxicity or mutation. We had previously tried using the *E. coli* strain KL740 (Coli Genetic Stock Center (CGSC)# 4382) with a temperature-sensitive lambda phage repressor, but had been able to obtain successful function. Thus, in an effort to create a successful clone to grow our plasmid, we switched to another canonical strong constitutive promoter, J23101 from the Anderson collection. While the strengths of the promoters are not identical, we were impressed with our system’s ability to compensate for promoter strength during our calibration process.

- Page 4: why did the authors not push the reporter plasmid concentrations higher? They are clearly not at a local optimum yet, though they may be for the sensor module. Could that perhaps push sensitivity higher or improve other sensor parameters? Would it change whether or not the sensor module has a plateau point, or where it is?

Answer: While the reporter plasmid concentrations did not reach a maximum, the responses at 30 nM and 100 nM concentrations were similar, particularly at higher benzoic acid concentrations. Additionally, at a concentration of 100 nM, the DNA plasmids being added have a noticeably higher viscosity. As cell-free protein synthesis reactions are influenced by molecular crowding, much higher levels of reporter DNA could exhibit effects on the reaction decoupled from the reporters themselves. Thus, to avoid complicating our system further with this aspect, we chose to leave 100 nM as the maximum DNA concentration we considered. However, if future sensors still showed high variability at these concentrations, that would indeed be an interesting avenue to pursue.

- Page 5: The authors allude to greater sensitivity than previous work but provide no quantitative comparison.

Answer: We have removed the non-quantitative sensitivity comparison to the work by **Libis et al.**, as that publication did not directly quantify sensitivity, and instead highlight the cell-free sensor's faster response time and larger fold change and dynamic range. See also response to Reviewer #1.

- Page 5: The authors' language here is too strong. They claim to "conclude" and "identify" and prove things, but given their scope of work they have not done that. They have only generated some modest hypotheses, and not tested any of them.

Answer: We have altered the wording of our model assessment as follows: "Based on these models, we hypothesized that the observed bell-shaped response is likely due to competition between the different modules, leading to an important and unnecessary enzyme production at high DNA concentrations that divert resources such as RNA polymerase, ribosomes, and energy from sfGFP transcription and translation, as well as generating toxic byproducts. Moreover, we provide evidence that the shifting peak between the two setups is most likely due to lower expression of CocE (**Supplementary Text** and **Supplementary Figure 4**)." Additionally, due to the modifications we have made to the model formulation, as well as the testing the shift in optimal metabolic enzyme concentration for a different set of transcription factor and reporter DNA concentrations experimentally, we feel that we have strengthened the usefulness and validity of the model to warrant the updated language.

- Page 7 first paragraph, the authors seem to suggest that the low cross-reactivity will be generalizable. This is very likely specific to their work. Their two targets are very, very far away in chemical space from benzoic acid. The only thing they have in common is the aromatic ring. The targets have huge functional groups that could obviously be the cause of that specificity for the transcription factor, but other targets may not.

Answer: This is not necessary specific to our work, but we also acknowledge that crosstalk might occur, particularly if the product is highly similar to the substrate. We added a sentence to clarify the need for using a transcription factor that responds to the final product and not the initial substrate.

- The authors appear to have changed scales for their graphs? This greatly, greatly complicates interpretation of the data. Please keep them constant, especially when claims are made (as the authors do) about the quantitative results being the same across the different sensors.

Answer: There are different reasons for changing scales: the timescale of the kinetics was changed for CocE experiments because kinetics are longer at 30°C. Regarding the HipO vs CocE transducers titrations, both sensors show different fluorescence intensity values (Fig3A),

probably because of the different experimental conditions. However, fold changes are clearly similar. For that reason we chose to use different scales to really emphasize to the reader the minimum and maximum responses of each circuit. We did not intend to compare them in Fig 3A in terms of intensity, but only to determine the optimal DNA concentrations for “peak” activity. Indeed, a comparison is provided in terms of fold change (more relevant) in Fig 3B.

- Page 7, the authors claim to provide a “rigorous scheme to optimize cell-free biosensor performance”; no such novel rigorous scheme is provided. They just tested a bunch of concentrations, as any engineer might have done.

Answer: This point was also raised by reviewer #1. Part of the novelty of our cell-free system optimization procedure was the ability to modularize circuit components, individually titrate each DNA component to maximize signal fold change and dynamic range in the face of cell-free, resource-limited environment.

We did not merely “tested a bunch” of concentrations, there was actually a rationale on how the concentrations were tested. We started determining the optimal DNA concentrations for TF and reporter plasmids, and then moved to identifying best enzyme DNA concentrations, testing DNA concentrations over several order of magnitudes.

To highlight this aspect, the above sentence has been changed to: “In addition, we provide a three order-of-magnitude titration for each DNA component to optimize cell-free biosensor performance along with a mathematical model enabling a better understanding of the parameters governing cell-free biosensors response which will help future optimisation of such devices.”

- Page 7, the authors claim the model will help future development. How? See above analyses.

Answer: Given the efficiency of the enzymes (low amounts of enzyme DNA are required for the biosensors to work), the model suggests that the optimal transcription factor/reporter concentrations chosen will remain the optimal ones despite adding enzyme DNA. However, on less efficient enzymes, this equilibrium could change and we provide the model and scripts to analyse these behaviors given new experimental data trying to implement our methods. We also show that our model can be used to computationally explore several parameter ranges, which would be tedious to do experimentally (see earlier response). Based on the the model, we performed a new experiment in the revised manuscript and observed the expected shift of the curve at higher BenR DNA concentrations. We believe this provide a case for using model to explore the parameter space and suggest interesting experimental conditions to test. In all, we agree that the model will need improvement in the future, but it is a starting point that incorporates novel relevant parameters (i.e. plasmid DNA concentration of different circuit components) from which we and other can build.

- Page 7, the authors say they have 64 biomarkers but the supplement says general metabolites. The supplementary file includes molecules like carbon dioxide and hydroxide, suggesting it is not just biomarkers.

Answer: We replaced biomarkers by metabolites in the main text.

- Page 7, additional context should be given to those biomarkers. If a disease diagnosis requires 20 markers to be measured, is this much different than running an existing clinical panel or using analytical chemistry techniques like mass spectrometry?

Answer: We do not know if a disease will require 20 markers to be detected to be properly diagnose, but if it was the case, then the biosensing technology (if successful) would still be cheaper to develop, easier to use, and more portable than running a panel of analytical assays or mass spectrometry.

- Page 11: Why did they “change” the promoter for CocE? Justification for an arbitrary decision like that is needed, especially when the authors then later blame this decision in part for problems in their model development.

Answer: As mentioned in a previous response, during cloning we found that using the same Pr-OR2-OR1 promoter as the BenR and HipO plasmids for CocE led to lethal toxicity. We had previously tried using an *E. coli* strain KL740 (Coli Genetic Stock Center (CGSC)# 4382) with a temperature-sensitive lambda phage repressor, but had been able to obtain successful function. Thus, in an effort to create a successful clone to grow our plasmid, we switched to another canonical strong constitutive promoter, J23101 from the Anderson collection. While the strengths of the promoters are not identical, we were impressed with our system’s ability to compensate for promoter strength during our calibration process and additionally ran tests of the two constitutive promoters to judge their relative strengths for tuning of our model.

- Is a Hill equation really well-justified? Where would the cooperativity come in? This assumption should be justified better before adding another adjustable parameter.

Answer: The authors thank the reviewer for this remark. The Hill equation was replaced by a Michaelis-Menten equation with no cooperativity, which still explains the observed effects in the data. The biosensor which was previously modeled as a one step Hill function is now modeled as first activation of BenR and then activation of the promoter.

- The authors appear to have a typo with different terms K_{tx} and K_{tox} .

Answer: The typo has been corrected.

- The meaning of the gamma, pi, lambda, and delta terms should be made more evident

even in the absence of the detailed appendix. This needs to be digestible, especially when the authors claim that nothing needed to be fitted and all of the data came from the literature.

Answer: The text has been made more explicit regarding the terms in the listed equations. We are still referring the experienced reader interested in model development to the detailed appendix, as they believe it is as important to understand the development as much as the final equations when it comes to formalising biological knowledge. Moreover, a table was added in the supplementary text summing up the variables and their definitions.

- Page 20, the authors say they made the affinity “smaller”. Do they mean “weaker”? Because that would be quantitatively “bigger”, not “smaller”. (“greater”, not “lesser”, magnitude)

Answer: The affinity of the ribosome to the mRNA was smaller, which effectively translates to a higher dissociation constant. We thank the reviewer for his remark as it helped us clarify this part of our explanation.

- Page 20, shouldn't the impact of using a different plate reader be taken away by the normalization of fluorescent signals to a fixed concentration of strong promoter-driven GFP?

Answer: This normalisation should indeed take away the impact of using another plate reader. However, we do observe a difference in the scale response of our two enzymatic assays which our model currently does not account for. This could be due to the different temperatures at which the experiments were run, as we now discuss in the Supplementary Test with the following paragraph: “Moreover, the CocE experiment was performed at 30°C as it is the optimal temperature for this enzyme. Our modeling assumption was that this impacted only kinetic parameters, which is therefore included in our model. However, it might also affect the benzoic acid reporter which the model does not account for.”

- Page 21: The authors identified a shortcoming in their model (spontaneous transformation), and a potential solution and didn't bother to fix it? The justification here seems weak.

Answer: The model has been updated to address the reviewer's concern and now includes spontaneous transformation, which solves this issue. The main point of the model was still the understanding of the other effects linked mostly to resource competition and enzyme kinetics but this side effect is now accounted for.

- Why do both hippuric acid and cocaine have a larger baseline measurement with no enzyme at 500, 1000 uM than benzoic acid?

Answer: The slight signal at very large concentrations of hippuric acid and cocaine in the absence of metabolic enzyme are likely due to non-specific transcription factor activation (as

reporter by **Libis et al.**) and hydrolyzation, respectively.

- The authors seem to have gotten some parameter values from established databases of *in vitro* or *in vivo* (but not cell free) parameters, and others from published papers of *in vivo* work. Is there really a good reason to expect those to all just work perfectly like for hippuric acid? Or is there perhaps not much sensitivity once they fix a few parameters to their system?

Answer: The parameters showing the highest qualitative impact on the effects we were interested in (why increasing enzyme DNA leads to bell-shaped signal response and why the peak shifts between constructs) are the promoter and RBS affinity values, as well as obviously DNA concentration. These parameters were determinant in obtaining the bell-shaped signal response. Other parameters, that were obtained from enzyme databases, also had an effect on the exact shape of the bell but not its existence, or on the transducer values, especially for lower signals. Since the effects of those experimentally determined parameters could be compensated by changing promoter and RBS values that were qualitatively chosen on a relative scale, we decided to fix the enzyme kinetic parameters according to the databases and only vary the parameters that depended on our genetic construct. Therefore, fixing enzyme parameters according to literature and choosing other sensitive parameters on a relative scale with only slight tuning leads to a satisfying explicative model.

So there was not really a lot of sensitivity on parameters obtained for enzyme kinetic data as long as the enzyme is reasonably effective, which is why we expect these parameters to work with only slight tuning of the above-mentioned parameters. This would probably not be as accurate for a protein that folds poorly in cell-free extracts, as much lower efficiency than measured **in vitro** could be expected.

- Why did the authors divide rate parameters by 10? Though the step was well-disclosed, this reviewer finds some difficulty in finding the explicit justification.

Answer: Many publications in cell-free modeling state that there is an order of magnitude difference between cell-free and **in vivo** rates (added in the references section of the Appendix of this work). While more detailed rate changing could have been investigated, this was not done for three reasons. First, there is weak sensitivity of the results presented on parameters concerned by this change (production and degradation rates mostly). Second, the effects of finer rate adaptation to cell-free would be the same on all proteins involved (enzymes, transcription factor and reporter) and corrected by an adequate shift in affinity values, so this would not change the qualitative effects explained by the model as it would apply to all proteins. Third, given the two previous reasons, we believe simplicity was more important than detailed rate changing on those parameters. In any case, given the model formulation, this can easily be updated for later applications by parameter re-tuning if so decided. Our main aim was once again to obtain a qualitative understanding of the effects we mentioned, which would not be affected by more detailed rate tuning than the order of magnitude change we applied.

Reviewers' Comments:

Reviewer #1 (Remarks to the Author)

Voyvodic et al. here presents a revised manuscript. While this reviewer still questions the novelty of the rational engineering pipeline, the revised manuscript provides a highly sought-for demonstration of the applicability of the cell-free biosensors for chemical quantifications in complex samples. While there could be further biosensor optimisations to apply them for broad-range and simple screening, the proof-of-principle demonstrations for the sensors to work in "the field" are very solid.

This reviewer only has a few minor points to be corrected/commented on before the manuscript should be ready for publication in Nature Communication.

Minors:

L168: Replace "saw" with "observed"

L200: replace "'in a monitoring capacity" with "as a monitor"

L273-275: This can be removed. It is a repetition of the Figure 4d legend (L709-711).

L319-320: I think the authors mean higher sensitivity (or increased operational range) instead of dynamic range. A dynamic output range of 4.3-8.8x (L272) should be fine given the linear range of detection falls within application needs.

L450: I am not familiar with the regulations related to use of cocaine for research purposes, but authors should ensure that they comply with ethical standards and consider if info related to purchase of the cocaine inducer is to be inserted here.

L691: What is the negative control? Please indicate as not mentioned in Methods either.

L693: Insert "(dashed line)" following "cut-off point".

L697: Why perform calibration curve following 4 hours and monitoring of complex samples following 1 hr (L691). Please put a brief elaboration here or in L217-219.

Reviewer: Michael Krogh Jensen

Reviewer #3 (Remarks to the Author)

The authors have substantively responded to almost all of this reviewer's previous critiques. The authors have also substantially walked back many of the over-reaching and over-broad claims that previously made the data not support the conclusions of their paper. The addition of measurements in real-world complex samples is greatly, greatly appreciated, and makes the work much more interesting and convincing.

In short: this reviewer still isn't totally convinced that this is Nature Communications worthy, but it is much closer now, and it is a whole lot better in this revision. The fundamental question underlying publication here is: is the use of an enzyme, found by a previously-published algorithm, to turn one molecule into another that is detectable using a previously-published back end sensor truly a Nature Communications-type advance? Going back to the critique previously presented in this reviewer's last critique: glucose oxidase and other redox sensors are used widely for secondary detection. Even the tyrosine to melanin example from Santos and Stephanopoulos indicates that "turning one molecule into another that can easily be detected" is established in synthetic biology. So, is doing something that is already established, with a previously-published software piece and previously-published back-end sensor, truly Nature Communications worthy?

This reviewer would 100% recommend this version of the manuscript for publication in ACS Synthetic Biology without hesitation. And that comes from the implementation of the soda and urine detection. But as to whether the whole thing really has sufficient novelty... this reviewer is still not totally convinced. Not so astounded as to think that it is absurd to publish it in Nature

Communications, but if this reviewer had to imagine things that would truly get cited as being important in their area, this is not one that is easy to imagine would get lots of citations. Not that it wouldn't get any as a cell-free sensor in a complex sample, but it just doesn't strike one as quite the fundamental advance.

To address a couple of comments by the authors in their response:

Original contributions from our work include: 1) modularization of circuit components on different vectors; 2) combinatorial titration to identify optimal working concentrations; 3) optimization of our system in the context of resource depletion occurring in cell-free systems; 4) new applications for cell-free biosensors in the context of complex samples including beverages and clinical samples, without any pre-processing.

Points 1 and 2 are oversold significantly. Modularization of circuit components on different vectors is absolutely NOT novel to their work. This is common throughout cell-free work. Similarly, combinatorial titration is neither insightful nor unique. It is standard tuning, or design-build-testing. Point 3 has some modest contribution, and point 4 is a great demo.

The authors later reiterate their defense of the combinatorial testing:

We did not merely "tested a bunch" of concentrations, there was actually a rationale on how the concentrations were tested. We started determining the optimal DNA concentrations for TF and reporter plasmids, and then moved to identifying best enzyme DNA concentrations, testing DNA concentrations over several order of magnitudes.

Again, this is just standard testing and tuning that any bioengineer would have done. There is nothing novel at all there. Not that it is bad per se, there is no problem with it and it is the right thing to do, but claiming novelty and importance on that point is overselling it.

Other new things noticed in the new version:

Figure 4b: How does one decide if the negative drinks are non-zero? Would there be some kind of critical value for a concentration?

Why did the authors dilute for hippuric acid but not for cocaine (both in urine)? While the transparency is appreciated, there ought to be a little more clarity on why that decision was made.

The cocaine "physiological" range as described by the authors in the paper is high and generous. From that paper, the number they cite is an outlier, and other papers cite the common upper limit from a factor of two to a factor of ten lower (for example, PMC3128807), limiting potential clinical applicability.

The cocaine 2-fold change is also not quite impressive, especially given the variability across urine samples. This is not the degree of response one expects based on the rest of the paper, and highlights the importance of using the appropriate complex samples for testing. But nonetheless, it does appear for some concentrations to be statistically significant.

Supplementary Figure 14 appears to be missing in this reviewer's file.

Response to reviewers: “Plug-and-Play Metabolic Transducers Expand the Chemical Detection Space of Cell-Free Biosensors”

Peter L Voyvodic, Amir Pandi, Mathilde Koch, Ismael Conejero, Emmanuel Valjent, Philippe Courtet, Eric Renard, Jean-Loup Faulon*[†], and Jerome Bonnet*[†]

We would like to thank the reviewers for their constructive comments which we believe have improved the quality of the paper. Below is a full point-by-point response to the reviewer's comments. Our answers are in blue.

Reviewers' comments:

Reviewer #1 (Remarks to the Author):

Voyvodic et al. here presents a revised manuscript. While this reviewer still questions the novelty of the rational engineering pipeline, the revised manuscript provides a highly sought-for demonstration of the applicability of the cell-free biosensors for chemical quantifications in complex samples. While there could be further biosensor optimisations to apply them for broad-range and simple screening, the proof-of-principle demonstrations for the sensors to work in “the field” are very solid.

This reviewer only has a few minor points to be corrected/commented on before the manuscript should be ready for publication in Nature Communication.

Minors:

L168: Replace “saw” with “observed”

L200: replace “in a monitoring capacity” with “as a monitor”

L273-275: This can be removed. It is a repetition of the Figure 4d legend (L709-711).

Answer: We thank the reviewer for these corrections and have replaced or removed the text accordingly.

L319-320: I think the authors mean higher sensitivity (or increased operational range) instead of dynamic range. A dynamic output range of 4.3-8.8x (L272) should be fine given the linear range of detection falls within application needs.

Answer: Indeed the important part of increasing the dynamic range is expanding to a lower detection limit. To clarify this, the sentence has been reworded as: “...this application would benefit from achieving a lower detection limit, for example through the use of downstream genetic amplifiers”

L450: I am not familiar with the regulations related to use of cocaine for research purposes, but authors should ensure that they comply with ethical standards and consider if info related to purchase of the cocaine inducer is to be inserted here.

Answer: We would like to direct the reviewer to the Methods subsection 'Cell-free sensor optimization reactions' where we have included the following sentence after indication that cocaine hydrochloride was purchased from Sigma-Aldrich: "Permission to purchase cocaine hydrochloride was given by the French drug regulatory agency (Agence Nationale de Sécurité du Médicament et des Produits de Santé) to allow development of a new biosensor."

L691: What is the negative control? Please indicate as not mentioned in Methods either.

Answer: The negative control with water instead of commercial beverage has now been referenced in the figure legend and methods section.

L693: Insert "(dashed line)" following "cut-off point".

Answer: The text has been changed accordingly.

L697: Why perform calibration curve following 4 hours and monitoring of complex samples following 1 hr (L691). Please put a brief elaboration here or in L217-219.

Answer: The quantification measurements and calibration curve were both measured after four hours for Figure 4b. To clarify this, we have edited to sentence as follows: "Beverages were added at 1:10 dilution to cell-free reactions for four hours and the benzoic acid concentration was determined using a calibration curve (Supplemental Figure 10)."

Reviewer #3 (Remarks to the Author):

The authors have substantively responded to almost all of this reviewer's previous critiques. The authors have also substantially walked back many of the over-reaching and over-broad claims that previously made the data not support the conclusions of their paper. The addition of measurements in real-world complex samples is greatly, greatly appreciated, and makes the work much more interesting and convincing.

In short: this reviewer still isn't totally convinced that this is Nature Communications worthy, but it is much closer now, and it is a whole lot better in this revision. The fundamental question underlying publication here is: is the use of an enzyme, found by a previously-published algorithm, to turn one molecule into another that is detectable using a previously-published back end sensor truly a Nature Communications-type advance? Going back to the critique previously presented in this reviewer's last critique: glucose oxidase and other redox sensors are used widely for secondary detection. Even the tyrosine to melanin example from Santos and Stephanopoulos indicates that "turning one molecule into another that can easily be detected" is established in synthetic biology. So, is doing something that is already established, with a previously-published software piece and previously-published back-end sensor, truly Nature Communications worthy?

This reviewer would 100% recommend this version of the manuscript for publication in ACS Synthetic Biology without hesitation. And that comes from the implementation of the soda and urine detection. But as to whether the whole thing really has sufficient novelty... this reviewer is still not totally convinced. Not so astounded as to think that it is absurd to publish it in Nature Communications, but if this reviewer had to imagine things that would truly get cited as being important in their area, this is not one that is easy to imagine would get lots of citations. Not that it wouldn't get any as a cell-free sensor in a complex sample, but it just doesn't strike one as quite the fundamental advance.

To address a couple of comments by the authors in their response:

Original contributions from our work include: 1) modularization of circuit components on different vectors; 2) combinatorial titration to identify optimal working concentrations; 3) optimization of our system in the context of resource depletion occurring in cell-free systems; 4) new applications for cell-free biosensors in the context of complex samples including beverages and clinical samples, without any pre-processing.

Points 1 and 2 are oversold significantly. Modularization of circuit components on different vectors is absolutely NOT novel to their work. This is common throughout cell-free work. Similarly, combinatorial titration is neither insightful nor unique. It is standard tuning, or design-build-test-ing. Point 3 has some modest contribution, and point 4 is a great demo.

The authors later reiterate their defense of the combinatorial testing:

We did not merely “tested a bunch” of concentrations, there was actually a rationale on how the concentrations were tested. We started determining the optimal DNA concentrations for TF and reporter plasmids, and then moved to identifying best enzyme DNA concentrations, testing DNA concentrations over several order of magnitudes.

Again, this is just standard testing and tuning that any bioengineer would have done. There is nothing novel at all there. Not that it is bad per se, there is no problem with it and it is the right thing to do, but claiming novelty and importance on that point is overselling it.

Other new things noticed in the new version:

Figure 4b: How does one decide if the negative drinks are non-zero? Would there be some kind of critical value for a concentration?

Answer: For our rapid, one-hour detection in Figure 4a, we deemed that drinks negative for benzoates if the fold change was less than 5 (shown as a dashed line on the graph). Depending on the application and expected levels of benzoates, this threshold could obviously be adjusted.

Why did the authors dilute for hippuric acid but not for cocaine (both in urine)? While the transparency is appreciated, there ought to be a little more clarity on why that decision was made.

Answer: As we wanted to quantify the endogenous levels of hippuric acid in urine, we opted for testing after dilution as: a) there would be little interference from the addition of urine in the cell-free reaction (Supplemental Figure 11) and b) we were able to detect signal even after dilution within the range of our calibration curve. For cocaine, we recognized in designing the experiment that the levels of cocaine we wanted to test would be near the limit of detection of our system. If we chose to add our cocaine spike to diluted urine, we would need to be able to detect a ten-fold lower signal to account for the dilution. Thus, we opted to use undiluted urine to detect the lowest level possible.

The cocaine “physiological” range as described by the authors in the paper is high and generous. From that paper, the number they cite is an outlier, and other papers cite the common upper limit from a factor of two to a factor of ten lower (for example, PMC3128807), limiting potential clinical applicability.

The cocaine 2-fold change is also not quite impressive, especially given the variability across urine samples. This is not the degree of response one expects based on the rest of the paper, and highlights the importance of using the appropriate complex samples for testing. But nonetheless, it does appear for some concentrations to be statistically significant.

Answer: We recognize that the value in the paper is higher than many of the others tested; however, there are several other patients that have cocaine concentrations which would register ~3-4x fold change in Figure 4d. In an effort to not oversell our claims, we decided to label Figure 4d with 'Previously Clinically Detected Concentrations' rather than something like 'Clinical Range' to illustrate this point. Additionally, we now mention in the discussion section that "...this application would benefit from achieving a lower detection limit, for example through the use of downstream genetic amplifiers"

Supplementary Figure 14 appears to be missing in this reviewer's file.

Answer: We apologize for the absence of the figure in pdf formatting. The figure is present in the final version.